# Diversification of heart progenitor cells by EGF signaling and differential modulation of ETS protein activity

Benjamin Schwarz, Dominik Hollfelder, Katharina Scharf, Leonie Hartmann, Ingolf Reim*

Department of Biology, Division of Developmental Biology, Friedrich-Alexander University of Erlangen-Nürnberg, Erlangen, Germany

**Abstract** For coordinated circulation, vertebrate and invertebrate hearts require stereotyped arrangements of diverse cell populations. This study explores the process of cardiac cell diversification in the *Drosophila* heart, focusing on the two major cardioblast subpopulations: generic working myocardial cells and inflow valve-forming ostial cardioblasts. By screening a large collection of randomly induced mutants, we identified several genes involved in cardiac patterning. Further analysis revealed an unexpected, specific requirement of EGF signaling for the specification of generic cardioblasts and a subset of pericardial cells. We demonstrate that the Tbx20 ortholog Midline acts as a direct target of the EGFR effector Pointed to repress ostial fates. Furthermore, we identified Edl/Mae, an antagonist of the ETS factor Pointed, as a novel cardiac regulator crucial for ostial cardioblast specification. Combining these findings, we propose a regulatory model in which the balance between activation of Pointed and its inhibition by Edl controls cardioblast subtype-specific gene expression.

DOI: https://doi.org/10.7554/eLife.32847.001

## Introduction

The heart consists of a variety of cells with distinct molecular and physiological properties in both vertebrates and invertebrates. A complex regulatory network of transcription factors and signaling pathways orchestrates the specification of these different cell populations and their proper arrangement within a regionalized working myocardium or other functional structures such as valves, inflow and outflow tracts (reviewed in *Greulich et al., 2011*; *Miquerol and Kelly, 2013*; *Rana et al., 2013*; for the invertebrate *Drosophila* heart see for example *Bodmer and Frasch, 2010*; *Lehmacher et al., 2012*; *Lovato and Cripps, 2016*; *Reim and Frasch, 2010*). For example, the vertebrate T-box gene *Tbx20* promotes working myocardial fate by restricting *Tbx2* expression and enabling the expression of chamber myocardium-specific genes (*Cai et al., 2005*; *Singh et al., 2005*; *Stennard et al., 2005*). By contrast, *Tbx2* and *Tbx3* repress working myocardium-specific gene expression and chamber differentiation in the non-chamber myocardium and thus contribute to the formation of endocardial cushions and structures of the conduction system (*Christoffels et al., 2004*; *Hoogaars et al., 2007*; *Singh et al., 2012*). Normal endocardial cushion formation also requires COUP-TFII, an orphan nuclear receptor transcription factor that regulates cell fate decisions in several tissues (*Lin et al., 2012*; *Wu et al., 2016*). In the embryonic mouse myocardium, COUP-TFII is restricted to atrial cardiomyocytes, a pattern consistent with a fate determination function that confers atrial over ventricular fate (*Lin et al., 2012*; *Wu et al., 2013*). This function appears to involve the up-regulation of *Tbx5* (*Wu et al., 2013*), another T-box gene with non-uniform cardiac expression and a fundamental role in heart development and human cardiac disease (*Basson et al., 1997*; *Bruneau et al., 1999*; *Bruneau et al., 2001*; *Ghosh et al., 2017*; *Steimle and*

*For correspondence:
ingolf.reim@fau.de

Competing interests: The authors declare that no competing interests exist.

**eLife digest** Organs contain many different kinds of cells, each specialised to perform a particular role. The fruit fly heart, for example, has two types of muscle cells: generic heart muscle cells and ostial heart muscle cells. The generic cells contract to force blood around the body, whilst the ostial cells form openings that allow blood to enter the heart. Though both types of cells carry the same genetic information, each uses a different combination of active genes to perform their role.

During development, the cells must decide whether to become generic or ostial. They obtain signals from other cells in and near the developing heart, and respond by turning genes on or off. The response uses proteins called transcription factors, which bind to regulatory portions of specific genes.

The sequence of signals and transcription factors that control the fate of developing heart muscle cells was not known. So Schwarz et al. examined the process using a technique called a mutagenesis screen. This involved triggering random genetic mutations and looking for flies with defects in their heart muscle cells. Matching the defects to the mutations revealed genes responsible for heart development.

Schwarz et al. found that for cells to develop into generic heart muscle cells, a signal called epidermal growth factor (EGF) switches on a transcription factor called Pointed in the cells. Pointed then turns on another transcription factor that switches off the genes for ostial cells. Conversely, ostial heart muscle cells develop when a protein called 'ETS-domain lacking' (Edl) interferes with Pointed, allowing the ostial genes to remain on. The balance between Pointed and Edl controls which type of heart cell each cell will become.

Many cells in other tissues in fruit flies also produce the Pointed and Edl proteins and respond to EGF signals. This means that this system may help to decide the fate of cells in other organs. The EGF signaling system is also present in other animals, including humans. Future work could reveal whether the same molecular decision making happens in our own hearts.
DOI: https://doi.org/10.7554/eLife.32847.002

*Moskowitz, 2017*). Furthermore, FGF-mediated receptor tyrosine kinase (RTK) signaling upstream of the cardiogenic transcription factor Nkx2-5 was recently shown to be required for the maintenance of ventricular chamber identity of cardiomyocytes in zebrafish (*Pradhan et al., 2017*). As emphasized below, spatial restriction of cardiac transcription factors as well as precisely controlled RTK signaling activities are not only important in vertebrate but also invertebrate hearts (*Gajewski et al., 2000*; *Lo and Frasch, 2001*; *Zaffran et al., 2006*; this work).

The *Drosophila* heart (dorsal vessel) comprises several types of cardiomyocytes (in the embryo called cardioblasts, CBs) and non-contractile pericardial cells (PCs) (*Bodmer and Frasch, 2010*; *Lovato and Cripps, 2016*). The progenitors of these cells are specified in segmentally repeated heart fields located at the intersection of BMP/Dpp and Wg/Wnt signaling activities (*Frasch, 1995*; *Reim and Frasch, 2005*; *Wu et al., 1995*). Subsequent specification of the definitive cardiogenic mesoderm depends on a conserved group of transcription factors, most importantly those encoded by the *Nkx2-5* ortholog *tinman* (*tin*), the *Gata4* ortholog *pannier* (*pnr*) and the *Dorsocross1-3* T-box genes (three *Tbx6*-related paralogs that also share features with *Tbx2/3/5*; in the following collectively called *Doc*) (*Alvarez et al., 2003*; *Azpiazu and Frasch, 1993*; *Bodmer, 1993*; *Gajewski et al., 1999*; *Junion et al., 2012*; *Reim and Frasch, 2005*; *Reim et al., 2003*; reviewed in *Reim and Frasch, 2010*; *Reim et al., 2017*).

While the identification of cardiogenic factors has greatly improved our understanding of early specification events, much less is known about the mechanisms that lead to the diversification of cardiac cell subpopulations. In this study, we mainly focus on the development of the two major cardioblast subpopulations: generic cardioblasts (gCBs), which build the main portion of the contractile tube ('working myocardium'), and ostial cardioblasts (oCBs), which form bi-cellular valves (ostia) for hemolymph inflow. Due to Hox gene inputs, ostial progenitor specification is limited to the abdominal region (*Lo et al., 2002*; *Lovato et al., 2002*; *Ponzielli et al., 2002*; *Ryan et al., 2005*; reviewed in *Monier et al., 2007*). Current research suggests that each abdominal hemisegment generates at

least seven distinct progenitors that give rise to six CBs (4 gCBs + 2 oCBs) and several types of PCs (Tin[+]/Even-skipped[Eve][+] EPCs, Tin[+] TPCs, and Odd-skipped[Odd][+] OPCs; *Bodmer and Frasch, 2010* and references therein). Whereas gCBs (a.k.a. Tin-CBs) maintain expression of *tin*, oCBs (a.k.a. Svp-CBs) specifically express the *COUP-TFII* ortholog *seven-up* (*svp*) and *Doc* (*Gajewski et al., 2000*; *Lo and Frasch, 2001*; *Ward and Skeath, 2000*; *Zaffran et al., 2006*). Previous work has shown that *Doc* is repressed in gCBs in a *tin*-dependent manner (*Zaffran et al., 2006*). Robust *tin* expression in turn depends on the *Tbx20* ortholog *midline* (*mid/nmr2*). The *mid* gene is first activated in gCB progenitors, but later, like its paralog *H15/nmr1*, becomes expressed in all cardioblasts (*Miskolczi-McCallum et al., 2005*; *Qian et al., 2005*; *Reim et al., 2005*). In oCBs, *svp* represses *tin* expression thereby permitting continued *Doc* expression in these cells (*Gajewski et al., 2000*; *Lo and Frasch, 2001*; *Zaffran et al., 2006*). In the abdomen, gCBs and most PCs are preceded by a precursor that undergoes symmetric division, whereas oCBs and half of the OPCs are derived from common, asymmetrically dividing CB/PC progenitors (*Alvarez et al., 2003*; *Han and Bodmer, 2003*; *Ward and Skeath, 2000*).

The process of progenitor specification in the somatic and cardiogenic mesoderm involves the antagonistic actions of RTK/Ras/MAPK and Delta/Notch signaling (*Carmena et al., 2002*; *Grigorian et al., 2011*; *Hartenstein et al., 1992*). Two types of RTKs, the fibroblast growth factor (FGF) receptor Heartless (Htl) and the epidermal growth factor (EGF) receptor EGFR, act positively on progenitor selection via MAPK signaling, although they are used by different progenitors to different extents (*Buff et al., 1998*; *Carmena et al., 2002*; *Michelson et al., 1998*). Htl and its FGF8-like ligands Pyramus (Pyr) and Thisbe (Ths) have a dual function as regulators of mesodermal cell migration and cell specification, with progenitors of the Eve[+] lineage as the most prominent example for the latter (reviewed in *Bae et al., 2012*; *Muha and Müller, 2013*). EGFR signaling appears to be dispensable for early mesoderm migration events (*Wilson et al., 2005*) but has been reported to contribute to the specification of particular cell types within the mesoderm, including subsets of adult muscle precursors (AMPs; *Figeac et al., 2010*) and the Eve[+] DA1 muscles (derived from the so-called P15 progenitors in the dorsal mesoderm; *Buff et al., 1998*; *Carmena et al., 1998*). By contrast, Eve[+]pericardial cells derived from the P2 progenitor were shown to form independent of EGFR activity. The exact contribution of EGFR signaling to *Drosophila* heart development has been less clear until now, but it was shown that EGFR loss-of-function results in a severe reduction of the numbers of cardioblasts, pericardial nephrocytes, and blood progenitors (*Grigorian et al., 2011*).

Molecularly, the predominant EGFR ligand in the embryo, Spitz (Spi), relies on the protease Rhomboid (encoded by *rho*) and the chaperon Star (S) for its conversion from a membrane-bound into its active form (reviewed in *Shilo, 2014*). In contrast to *spi*, *rho* expression is restricted to a limited number of cells in a complex and dynamic pattern, including cells of the cardiogenic area (*Bidet et al., 2003*; *Liu et al., 2006*), which points to *rho* expression being the most decisive factor for Spi-mediated EGFR activation. Among the most important downstream effectors of RTK/Ras/MAPK pathways are the ETS transcription factors PntP2 (encoded by *pointed/pnt*) and Yan/Aop (encoded by *anterior open/aop*). While PntP2 becomes an active transcriptional activator upon phosphorylation by MAPK, the transcriptional repressor Yan is negatively regulated by MAPK (*Gabay et al., 1996*; *O'Neill et al., 1994*). Unlike PntP2, a shorter isoform encoded by *pnt*, PntP1, is constitutively active but was shown to require activated MAPK for its transcriptional activation at least in some cell types (*Brunner et al., 1994*; *Gabay et al., 1996*; *Klämbt, 1993*; *O'Neill et al., 1994*). Notably, chordate Pnt orthologs (ETS1/2) were shown to contribute to cardiac progenitor formation in the tunicate *Ciona* and during transdifferentiation of human dermal fibroblasts into cardiac progenitors (*Davidson et al., 2006*; *Islas et al., 2012*). During early *Drosophila* cardiogenesis, Pnt favors expression of *eve* over that of another homeobox gene, *ladybird* (*lbe*, expressed in mesodermal cells immediately anterior of the Eve[+] cluster and later in TPCs and two of the four gCBs per hemisegment; *Jagla et al., 1997*) (*Liu et al., 2006*). In addition, Pnt promotes pericardial cell development and antagonizes CB fate, especially that of oCBs (*Alvarez et al., 2003*).

Despite the progress in the understanding of cardiac progenitor specification, the mechanisms that diversify progenitors of the oCB and gCB lineages have remained elusive. We have performed an unbiased large-scale mutagenesis screen to identify genes that regulate cardiac development in *Drosophila* embryos and found several mutants that show CB subtype-specific defects. On this basis, we discovered a novel and rather unexpected function of the EGF pathway in specifying the gCBs of the working myocardium, thus revealing an intimate link between cardioblast specification and

diversification. Furthermore, we identified ETS domain lacking (Edl a.k.a. Modulator of the activity of ETS, Mae) as a crucial regulator of the specification of inflow valve-forming oCBs. Edl possesses a SAM domain, which mediates binding to the SAM domain-containing ETS factors PntP2 and Yan, thereby inhibiting their activity as a transcriptional activator or repressor, respectively (*Baker et al., 2001*; *Qiao et al., 2006*; *Qiao et al., 2004*; *Tootle et al., 2003*; *Vivekanand et al., 2004*; *Yamada et al., 2003*). Our data imply that Edl enables *svp* expression and thus oCB fate by limiting the activity of PntP2, thereby blocking subsequent activation of important downstream targets such as *pntP1* and *mid*. Collectively, our data provide the basis for an elaborated model of cardiac cell fate diversification that links MAPK signaling, Pnt activity and the cell-type-specific expression patterns of key cardiac transcription factors.

## Results

### Novel EMS-induced mutants reveal a specific requirement of EGF signaling for the specification of generic cardioblasts

In order to identify genes involved in heart and muscle development in an unbiased manner, we have performed an EMS mutagenesis screen for chromosome two in *Drosophila melanogaster* embryos (*Hollfelder et al., 2014*). Several of the isolated mutants display a partial loss or irregular alignment of cardioblasts (CBs). Such defects may potentially result from mutations in genes that regulate the specification or differentiation of all CBs or only a particular CB subtype. In the latter case, disturbances in the characteristic '2 + 4' CB pattern of two ostial cardioblast (oCBs; Doc$^+$/Tin$^-$) and four generic CBs (gCBs; Doc$^-$/Tin$^+$) per hemisegment are to be expected. To analyze the cardiac pattern of mutants in more detail, we performed immunofluorescent double stainings for Doc and H15 (or alternatively Mef2) to label oCBs and all CBs, respectively. We then genetically and in part also molecularly mapped the mutations responsible for CB pattern anomalies (for details see the Materials and methods section and *Supplementary file 1*-Table S1). The class of mutants characterized by a loss of CBs contained several novel alleles of genes involved in RTK/Ras/MAPK signaling, which is consistent with the assumed role of this pathway in cardiac progenitor selection or maintenance (*Carmena et al., 2002*; *Grigorian et al., 2011*). However, no specific role for the specification of a particular cardioblast subtype or diversification of gCB versus oCB progenitors had been previously attributed to RTK/Ras/MAPK signaling. Our phenotypic analysis now shows that diminished EGF/EGFR but not FGF/Htl signaling leads to a preferential reduction of gCB numbers. Embryos with partially reduced FGF/Htl signaling, that is mutants lacking both copies of the FGF-encoding gene *pyr* and one copy of its paralog *ths*, as well as hypomorphic *htl* mutants, show an about equal reduction of gCB and oCB numbers (*Figure 1B*, for quantification see *Figure 1M*; additional examples in *Figure 1—figure supplement 1B,C*). This CB reduction can be explained by uneven spreading of the early mesoderm to Dpp-receiving areas. By contrast, several mutations mapped to EGF signaling components feature a preferential loss of gCBs. In strong *Egfr* mutants very few CBs can be found (*Figure 1C*, *Figure 1—figure supplement 1E*). Remarkably, the overwhelming majority of the residual CBs express Doc. The few remaining Doc-negative CBs are usually located toward the anterior and thus are possibly remnants of the oCB-free anterior aorta. In *spitz*, *rhomboid* and *Star* loss-of-function mutants, the number of Doc$^-$/Tin$^+$ CBs is strongly reduced while that of ostial Doc$^+$/Tin$^-$ CBs is nearly normal or in some cases even increased by a few cells (*Figure 1D–G,M*, *Figure 1—figure supplement 1F*, *Figure 1—figure supplement 2A–C*). In the wild type, the two pairs of sibling gCBs within each hemisegment can be further categorized as Lbe$^+$ (anterior pair) or Lbe$^-$ (posterior pair) subtypes. Since the above-mentioned *spitz* group mutants often feature a single pair of gCBs in each abdominal hemisegment, we tested whether these cells are preferentially Lbe$^+$ or Lbe$^-$, which would indicate that one of the two gCB progenitor types may be more sensitive to impaired EGF signaling. However, our finding that both types are about equally represented in *rho* mutants (*Figure 1—figure supplement 3*) argues against this assumption. Moreover, segment-by-segment analysis in homozygous *rho*$^{L68}$ mutants reveals that residual gCBs most frequently occur either as Lbe$^+$ or Lbe$^-$ pairs, whereas none of the analyzed residual gCB duplets consisted of a combination of both gCB types. This suggests that EGF function is required for the formation of gCB progenitors prior to their final division. Notably, progenitors of the oCB lineage apparently do not require activity of the ostial marker gene *svp* to develop and survive independently of EGF, since total CB

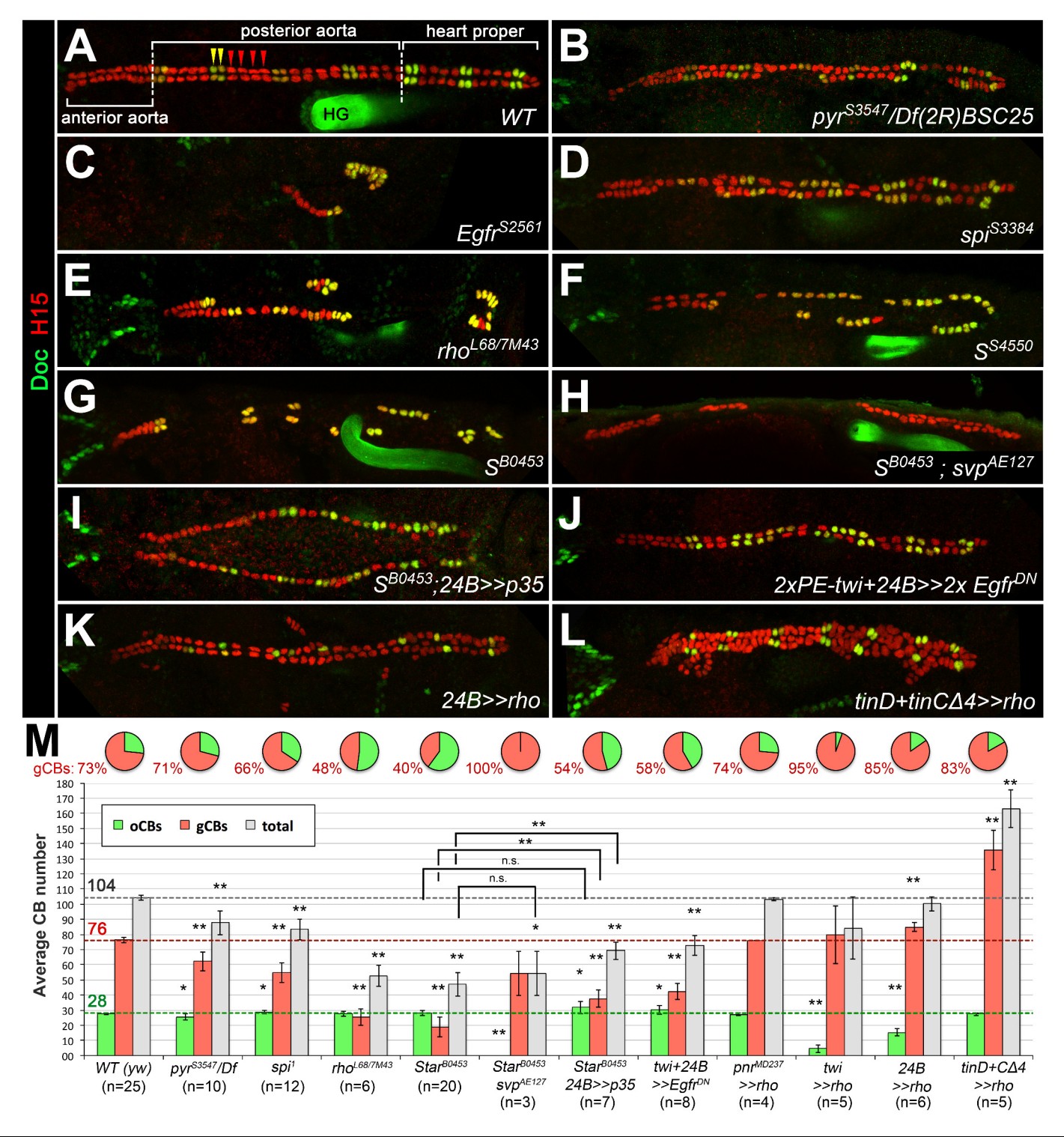

**Figure 1.** Genetic manipulation of EGF but not FGF signaling leads to cardioblast subtype-specific heart defects. Immunostaining for the cardioblast marker H15 (red) and the ostial cardioblast marker Dorsocross (anti-Doc2+3, green). (HG: hindgut with artificial staining in the lumen). All figures depict dorsal views of stage 16 embryos with anterior to the left unless noted otherwise. (**A**) Wild type (*WT*) CB pattern with regular alternation of gCBs (red) and oCBs (yellow) in the posterior aorta and the heart proper. The anterior aorta consists entirely of Doc⁻ CBs. (**B**) Mutant with reduced FGF activity (*pyr^S3547* over a deficiency, *Df(2R)BSC25*, that removes *pyr* and *ths*) showing a reduction of both CB types. (**C**) Homozygous *Egfr^S2561* mutant with a severe loss of CBs. Almost all remaining CBs are Doc⁺. Predominant reduction of gCBs is also observed in the EGF pathway-impairing *spitz* group mutants *spi^S3384* (**D**), *rho^7M43/rho^L68* (**E**), *S^S4550* (**F**) and *S^B0453* (**G**, showing an extreme case in which all retained CBs except for those of the anterior aorta

*Figure 1 continued on next page*

*Figure 1 continued*

are Doc⁺). (H) In $S^{B0453}$ $svp^{AE127}$ double mutants, total CB numbers are similar to that of $S$ single mutants, even though all CBs are Doc-negative. (I) If the apoptosis inhibitor p35 is artificially expressed in the mesoderm of $S$ mutants a mild increase in the number of CBs can be observed. Compared to the wild type, more Doc⁺ CBs are present. (J) Pan-mesodermal overexpression of dominant-negative *Egfr* results in a phenotype similar to *spitz* group mutants. Expression of *rho* in the entire mesoderm via *how*$^{24B}$-*GAL4* (K) or at later time in dorsal mesoderm cells via *tinD +tinCΔ4-GAL4* (L) generates supernumerary gCBs. By contrast, oCB specification is either reduced (K) or unaffected (L) in these backgrounds. (M) Quantification of Doc⁺ oCBs (green), Doc⁻ gCBs (red) and total cardioblasts (grey). The column bar chart depicts average numbers with standard deviation error bars. Asterisks indicate significant differences compared to the *y w* control (*WT*) assessed by Student's t-test (two-tailed, type 3; *=p < 0.05, **=p < 0.001; n.s. = not significant). Brackets indicate comparisons between other genotypes. Pie charts display the corresponding average fraction of oCBs and gCBs.

DOI: https://doi.org/10.7554/eLife.32847.003

The following source data and figure supplements are available for figure 1:

**Source data 1.** Quantification of Doc⁺oCBs, Doc⁻ gCBs and total cardioblasts.

DOI: https://doi.org/10.7554/eLife.32847.008

**Figure supplement 1.** Cardiac patterning phenotypes in additional alleles of FGF and EGF pathway mutants.

DOI: https://doi.org/10.7554/eLife.32847.004

**Figure supplement 2.** Extended analysis of cardiac patterning confirming the loss of Tin⁺cardiac cells in EGF pathway mutants.

DOI: https://doi.org/10.7554/eLife.32847.005

**Figure supplement 3.** Lbe⁺ and Lbe⁻ subtypes of generic cardioblasts are equally affected in EGF signaling mutants.

DOI: https://doi.org/10.7554/eLife.32847.006

**Figure supplement 4.** Analysis of apoptosis in *Star* mutants.

DOI: https://doi.org/10.7554/eLife.32847.007

numbers are similar in *Star* single and *Star svp* double mutants (compare *Figure 1H–1G*; quantification in *Figure 1M*).

Previous studies in EGF pathway mutants suggested that incorrectly specified mesodermal progenitors undergo apoptosis (*Buff et al., 1998*; *Grigorian et al., 2011*). Using TUNEL and anti-activated caspase stainings, we could not reliably detect signs of apoptosis in the Tin- or Doc-labeled cardiogenic mesoderm of *Star* mutants, while numerous signals were observed in other tissues (*Figure 1—figure supplement 4* and data not shown). Nevertheless, we obtained indirect evidence for the occurrence of at least some apoptosis by using the baculoviral apoptosis inhibitor p35 (*Zhou et al., 1997*). If p35 is artificially expressed in the mesoderm of *S* mutants the number of CBs slightly increases in comparison to *S* mutants without p35 (*Figure 1I,M*). Although this is consistent with a pro-survival function of EGF signaling, it does not fully account for the gCBs missing in *S* mutants. Of note, we detect a small, but statistically significant increase in the average number of Doc⁺ CBs in comparison to the wild type in *spi* mutants, in p35-expressing *S* mutants as well as in embryos overexpressing dominant-negative EGFR (*Figure 1M*), which suggests that at least some presumptive gCB progenitors adopt oCB-like fates at reduced EGFR activity. However, the observed effects are small and additional explanations such as persistence in an uncommitted dorsal mesoderm cell pool must be considered to fully explain the fate of all lost gCB progenitors (see discussion). Collectively, the cardiac patterning phenotypes imply that EGF signaling plays a major role in the correct specification of gCB progenitors, although we cannot exclude an additional function in cardiac cell survival that might be difficult to detect by the applied methods.

## Generic CBs and a subset of Odd⁺pericardial cells require spatially and temporally coordinated EGF signals

Because EGF signaling is involved in multiple processes during embryogenesis we next asked whether its impact on gCB specification is directly linked to signaling activity within mesoderm cells. Indeed, mesoderm-specific attenuation of the pathway by expression of a dominant-negative EGFR variant resulted in essentially the same phenotype as with the *spitz* group mutants (*Figure 1J,M*). Activation of the EGF pathway in mesoderm cells appears to be largely controlled by the spatially restricted expression of *rho* (*Bidet et al., 2003*; *Grigorian et al., 2011*; *Halfon et al., 2000*). Overexpression of *rho* with the pan-mesodermal *how*$^{24B}$-*GAL4* driver has been previously reported to affect the number of *tin*-expressing pericardial cells (*Bidet et al., 2003*), but CBs and their subtypes were not unambiguously labeled in these experiments. We extended these experiments using also other drivers. Consistent with a mesoderm-autonomous function, overexpression of *rho* in the dorsal

ectoderm (via *pnr*<sup>*MD237*</sup>-GAL4) has no significant effect on CB number or pattern (*Figure 1M* and data not shown). By contrast, all mesodermal *rho* overexpression setups increase the gCBs:oCBs ratio in comparison to the wild type (*Figure 1K–M* and data not shown). The impact on the absolute CB numbers depends on the timing and strength of transgene expression. The later *rho* is activated in mesodermal cells (with following drivers according to their temporal order and progressive spatial restriction: *twist-GAL4*, *how*<sup>*24B*</sup>*-GAL4* and *tinD +tinCΔ4-GAL4*) the larger the total number of CBs (*Figure 1K–M* and data not shown). This implies that *rho* activity needs to be tightly regulated, spatially as well as temporally. In the wild-type mesoderm, *rho* expression is first seen in the Eve<sup>+</sup> progenitor P2 (*Buff et al., 1998*; *Carmena et al., 1995*; *Halfon et al., 2000*) followed by expression in the adjacent CB progenitor-containing clusters C14 and C16 (*Bidet et al., 2003*; *Grigorian et al., 2011*; see also *Figure 2A–D*). Of note, stage 11 *rho* expression is still robustly observed in all C14/C16 clusters in *S* mutants (*Figure 2E* cf. 2A), showing that earlier patterning events are not disrupted in this situation. Later during stage 12, when *rho* RNA is normally found in developing CBs along the dorsal mesoderm margin, a reduction of *rho* expressing cells is apparent in *S* mutants (*Figure 2F* cf. 2C), which is consistent with defects in CB progenitor formation. Importantly, detection of active diphospho-MAPK is severely reduced in cardiac cells of *S* mutants already in the cardiogenic clusters at stage 11 as well as during 12 in which dpMAPK is normally detected in both ostial and generic CB progenitors (*Figure 2H,J* cf. 2G,I; later activity in cardiac cells appears to be less affected; *Figure 2L* cf. 2K). Similar observations were made for embryos with pan-mesodermal overexpression of the dominant-negative EGFR (data not shown). Altogether, this demonstrates that EGF signaling serves as the major positive input for MAPK activation during early gCB progenitor formation, whereas input from FGFs may gain importance in developing CBs at later stages for CB fate maintenance as was proposed previously (*Grigorian et al., 2011*).

Since half of the *odd*-expressing pericardial cells (OPCs) are siblings of oCBs, we also analyzed PCs in EGF-related mutants by Odd/Eve as well as Odd/Zfh1 double-stainings (*Figure 3A–C,E*; *Figure 3—figure supplement 1A–D* and data not shown). Consistent with the results of previous studies on Eve<sup>+</sup> progenitor derivatives (*Buff et al., 1998*; *Carmena et al., 2002*; *Su et al., 1999*), we detected EPCs in almost normal numbers in *spi* group mutants and in embryos with pan-mesodermal dominant-negative EGFR, whereas *spi*-dependent Eve<sup>+</sup> DA1 muscles were largely absent (*Figure 3B,C,E*). OPCs are strongly reduced in these loss-of-function backgrounds. Our quantification revealed that about half of the OPCs were lost in *rho*<sup>*7M43/L68*</sup> and other EGF pathway mutants (*Figure 3B,C,E*). A converse phenotype with many extra OPCs as well as Tin<sup>+</sup> PCs (TPCs, excluding the unaffected EPCs) is generated by *rho* overexpression with *tinD +tinCΔ4-GAL4* (*Figure 3D,E*; *Figure 3—figure supplement 1F*). Notably, the number of oCB-sibling OPCs (as identified by *svp-lacZ* reporter analysis) is not significantly reduced in *Star* mutants if compared to the wild type (*Figure 3F,G*), thus implying that the EGF signaling-dependent OPCs are those derived from symmetrically dividing OPC progenitors.

In sum, these data demonstrate that EGF pathway activity is required in the mesoderm specifically for the specification of the symmetrically dividing gCB and OPCs progenitors (and probably also for those of the TPCs, which we did not quantify in detail) but is largely dispensable or even detrimental for the specification of the *svp*-expressing oCB/OPC progenitors.

## The SAM domain protein Edl promotes specification of ostial cardioblasts by blocking Pointed activity

Our EMS screen also yielded mutants in which the number of ostial cardioblasts was specifically reduced. One such complementation group consisting of three alleles was mapped to the *numb* gene (alleles listed in *Supplementary file 1*-Table S1), which is consistent with its well-known function as a Notch suppressor during asymmetric cell division in the oCB lineage (*Gajewski et al., 2000*; *Ward and Skeath, 2000*). Preferential reduction of oCBs was also observed in the mutant line *S0520*. We found that its cardiac phenotype was caused by loss of the gene *ETS domain lacking* (*edl*) as part of a multi-gene deletion and named this mutant *Df(2R)edl-S0520* (*Figure 4A*, *Supplementary file 2*-Table S2). We identified *edl* as the gene responsible for the oCB losses by obtaining phenocopies with other *edl* mutants (*Figure 4A–D* and data not shown). The *lacZ* enhancer trap insertion allele *edl*<sup>*k06602*</sup> was used in most *edl* loss-of-function experiments since its cardiac phenotype is indistinguishable from that of *Df(2R)edl-S0520* and *Df(2R)edl-L19* (*Figure 4C,D* and data not shown), and we detected in this strain a small deletion that specifically destroys the *edl*

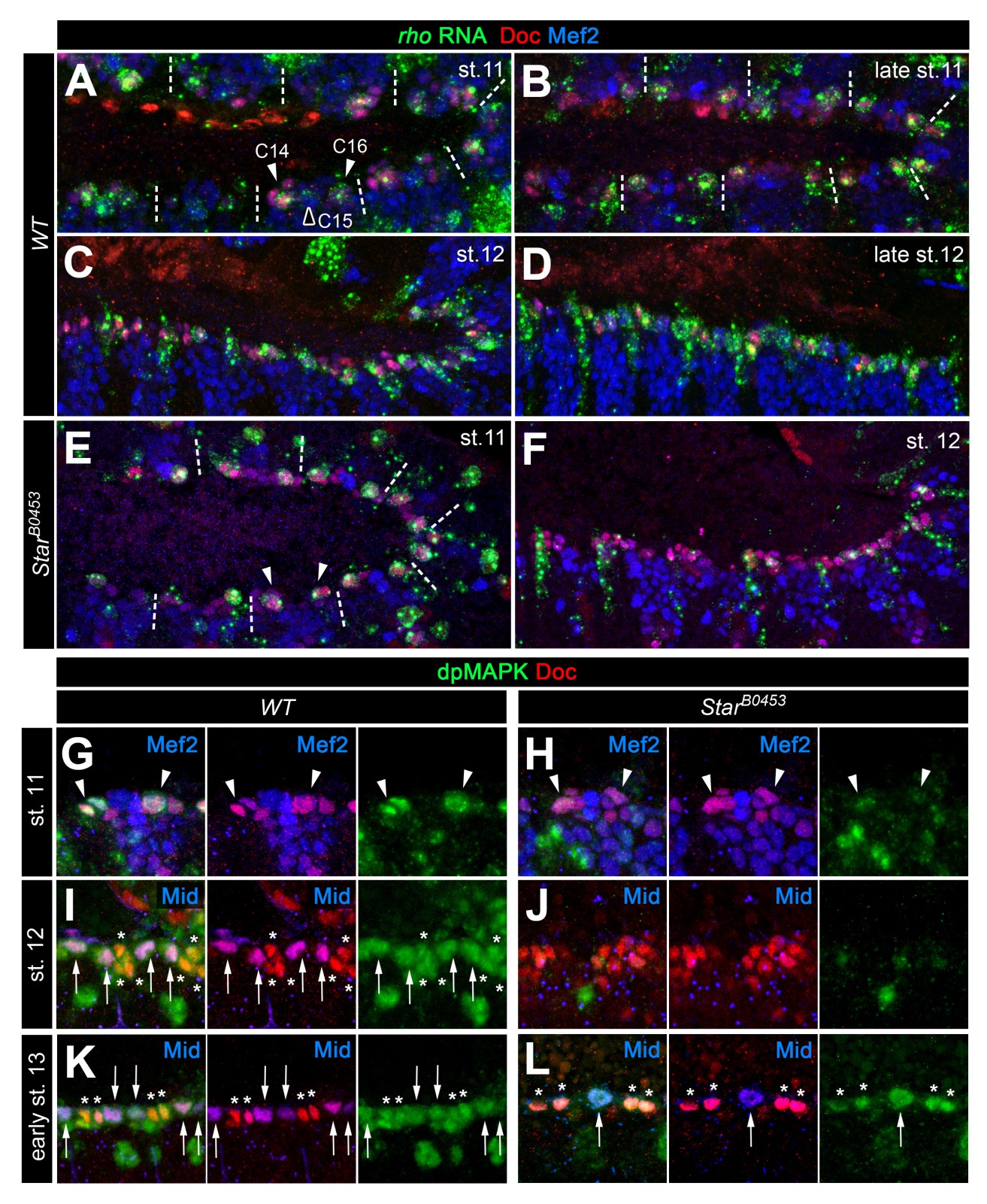

**Figure 2.** Expression of *rho* and MAPK activity in cardiac cells. (A–F) Detection of *rho* mRNA (green), Mef2 (blue) and Doc (red). (A) At stage 11, *rho* is detectable in clusters C14/C16 of the cardiac mesoderm (arrowheads) and is fading from the central Doc-negative region containing EPC and somatic muscle progenitors (empty arrowhead). Dashes separate units derived from adjacent mesoderm segments. (B) At late stage 11, *rho* is expressed at high levels in at least one cardiac progenitor per cluster close to the dorsal mesoderm segment borders. (C, D) As cardioblasts align near the dorsal

*Figure 2 continued on next page*

*Figure 2 continued*

mesoderm margin during stage 12, *rho* continues to be expressed in most CBs. (E,F) Detection of *rho* RNA in $S^{B0453}$ mutants showing normal *rho* expression in cardiogenic clusters at stage 11 (E, compare to A) and reduced cardiac expression at stage 12 (F, compare to C). (G–L) Detection of activated MAPK in the cardiogenic region of wild type (G,I,K) and $S^{B0453}$ mutant (H,J,L) embryos in immunostainings against diphospho-MAPK (dpMAPK, green), Doc (red) and either Mef2 or Mid (blue) as indicated in each panel. (G) dpMAPK is detectable in the Doc$^+$ cardiogenic clusters (arrowheads) of a stage 11 wild-type embryo. (H) This dpMAPK activity is severely reduced in *Star* mutants. (I) At stage 12, dpMAPK activity is observed in the Mid-expressing gCB progenitors (arrows) and in the Mid-negative oCBs and their sibling PCs (asterisks). (J) By contrast, both Mid and dpMAPK are severely reduced in stage 12 *Star* mutants. (K) Early stage 13 embryo after germ band retraction but prior to completion of the final mitotic division of the Mid$^+$ gCB progenitors. dpMAPK is still active in all cardiac cells (oCBs and gCBs labeled as in I). (L) In contrast to earlier stages, dpMAPK staining is prominently observed in both oCBs (asterisks) and the few formed Mid$^+$Doc$^-$ gCB progenitors (arrow) of *Star* mutants at the onset of stage 13.

DOI: https://doi.org/10.7554/eLife.32847.009

gene (*Figure 4A*, *Supplementary file 2*-Table S2). Furthermore, we were able to rescue the cardiac phenotype of *edl* by introducing a genomic *edl* transgene (*Yamada et al., 2003*; *Figure 4E*). Phenotypic rescue was also achieved, albeit with lesser efficiency, by artificially expressing *edl* in the dorsal mesoderm cells or in cardioblasts using the drivers *tinD-GAL4* and *tinCΔ4-GAL4*, respectively (*Figure 4F,G*), demonstrating that Edl is required directly within these cell types. In accordance, *edl* mRNA is found within the cardiogenic region during stages 10 to 12 (*Figure 4—figure supplement 1A–C*; *Figure 4—figure supplement 2A–D*), including prominent expression in early *svp*-expressing oCB progenitors (*Figure 4—figure supplement 2E*). Thereafter *edl* expression shifts to the pericardial region, where it persists until stage 15 (*Figure 4—figure supplement 1D* and data not shown).

A distinctive feature of *edl* mutants is that the normal '2 + 4' pattern of 2 Doc$^+$ CBs + 4 Doc$^-$ CBs is often transformed into a '1 + 5' pattern (e.g. bracket in *Figure 4D*), indicating a fate switch from ostial to generic CBs. However, Edl is not a direct activator of *Doc* expression because Doc is found in CBs of *edl* double mutants with CB-specific ablation of *tin* (*Figure 4I*), a phenotype reminiscent of that of CB-specific *tin* single mutants (*Figure 4H*; *Zaffran et al., 2006*). This suggests that *edl* normally contributes to the activation of *Doc* in oCBs via suppression of *tin*. This role of *edl* in CB patterning is further supported by the observation of some CBs with low levels of both Tin and Doc in *edl* mutants (*Figure 4K*; compare to the strictly complementary distribution of Doc and Tin in the wild type, *Figure 4J*).

Next, we analyzed Edl function by ectopic expression. Consistent with a mesoderm-autonomous function, overexpressing *edl* in the dorsal ectoderm via *pnr$^{MD237}$-GAL4* has no significant effect on cardiogenesis (data not shown). By contrast, overexpression of *edl* in the entire mesoderm via *twist-GAL4* results in an increase of CB numbers (*Figure 5A*) and a decrease of OPCs (described in the next subsection). The increase in Doc$^+$ CBs is disproportionately high. The extra Doc$^+$ CBs in the heart proper also activate ostial cell differentiation markers such as *wg* (data not shown). In agreement with the proposed function of Edl as a negative regulator of PntP2 (*Yamada et al., 2003*), our overexpression phenotypes of *edl* are very reminiscent to that of *pntP2*-specific mutants (*pnt$^{RR112}$* reported in *Alvarez et al. (2003)*; and *pnt$^{MI03880}$* shown in *Figure 5B*) and amorphic *pnt* mutants (*pnt$^{Δ88}$*, *pnt$^2$*; see *Figure 5E,I* and *Alvarez et al., 2003*). Accordingly, overexpression of constitutively active PntP2$^{VP16}$ (*Figure 5C*) or PntP1 (not shown) via *tinD +tinCΔ4-GAL4* causes a phenotype similar to that of *edl* loss-of-function mutants (*Figure 4C,D*). By contrast, analogous overexpression of the potential Edl target Yan/Aop leads to a loss of heart cells irrespective of their subtype (*Figure 5D*). These losses may result from a more general block in cell specification and differentiation since Yan has been related to such functions in several other types of MAPK-dependent progenitors (*Bidet et al., 2003*; *Caviglia and Luschnig, 2013*; *Halfon et al., 2000*; *Rebay and Rubin, 1995*). If the predominant function of Edl during CB specification is the inhibition of Pnt, *edl pnt* double mutants should mimic *pnt* mutants. In principle, this is what we observed (*Figure 5E,F*; quantifications in *Figure 5I*). By contrast, *edl aop* double mutants show an additive combination of *aop* and *edl* single mutant phenotypes (compare *Figure 5H* with 5G and 4D; see also quantifications in *Figure 5I*). Amorphic *aop* mutants display a reduction in CB number irrespective of CB subtype, which we ascribe to a permissive function during CB development that is probably linked to its well-documented role in restricting *eve* expression in the early dorsal mesoderm (*Bidet et al., 2003*; *Halfon et al., 2000*; *Liu et al., 2006*; *Webber et al., 2013*). Importantly, and in contrast to *edl* and *pnt* activity changes, manipulating *aop* activities does not lead to significant shifts in the oCBs:gCBs

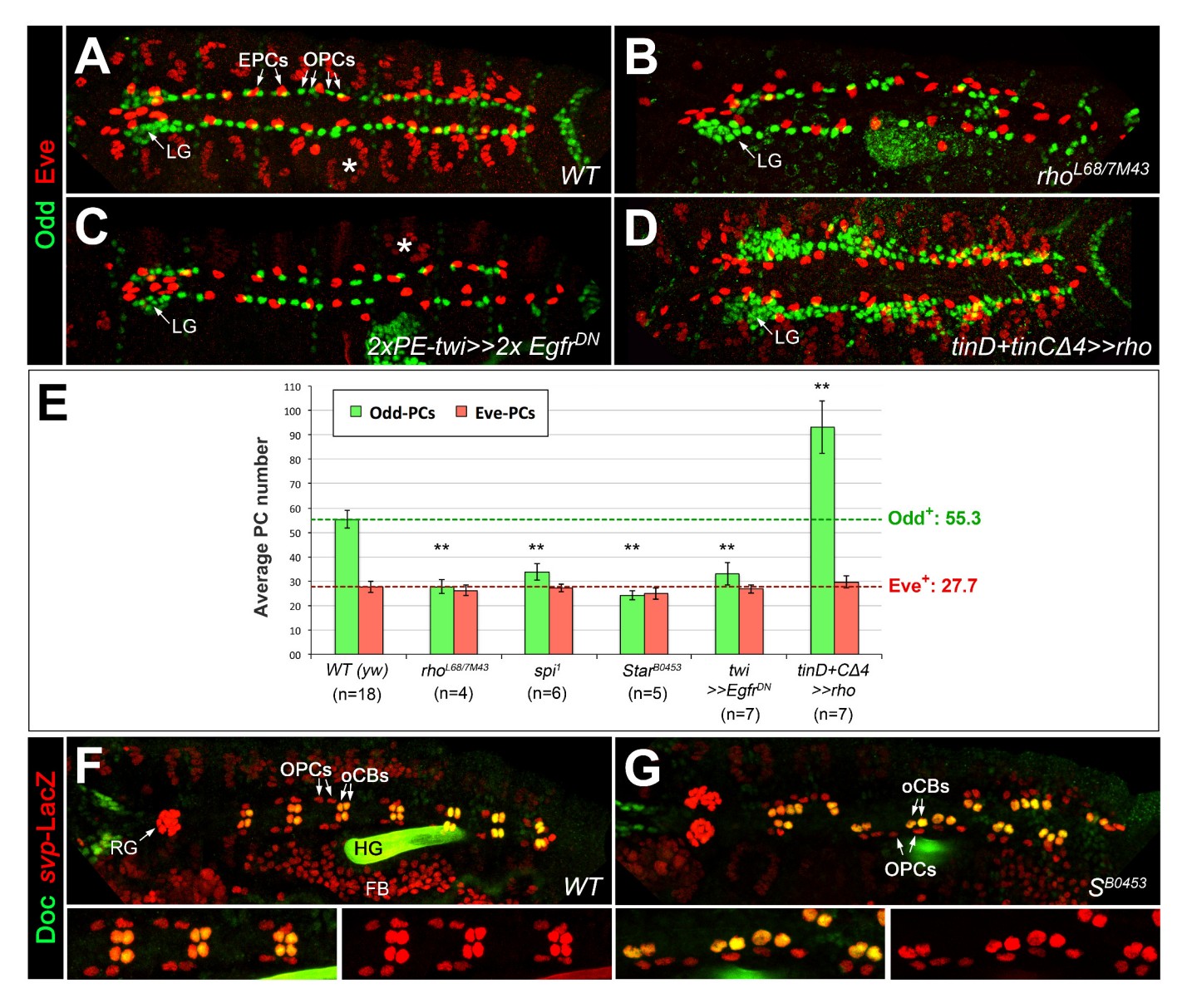

**Figure 3.** EGF signaling promotes the formation of Odd+PCs. (A–D) Odd/Eve staining to analyze pericardial cells (PCs). (A) In the wild type, each hemisegment contains four OPCs, two EPCs and one Eve+ somatic muscle DA1 (*). (B) Amorphic *rho7M43/L68* mutant with a loss of about half of all OPCs and all DA1 muscles. (C) Pan-mesodermal overexpression of the dominant-negative *Egfr* results in a phenotype similar to *rho* mutants. (D) Overexpression of *rho* in the dorsal mesoderm generates supernumerary OPCs. The number of EPCs is not affected by altered levels of EGF signaling. (E) Quantification of OPCs (green) and EPCs (red). Only abdominal PCs (located posterior to the lymph gland, LG) were included into the analysis. Significant differences compared to the *y w* control (*WT*) are designated as in *Figure 1*. Colored dashed lines mark the average numbers of OPCs and EPCs counted in the wild type. (F,G) Doc2+3/β-galactosidase (LacZ) staining in wild type (F) and *Star* mutant embryos (G) carrying a heterozygous copy of *svpAE127-lacZ* and showing presence of normal numbers of oCBs (Doc+/LacZ+) and their OPC siblings (Doc-/LacZ+). Bottom panels show a higher magnification and β-galactosidase single channel view of the upper panel. RG: ring gland, FB: fat body.

DOI: https://doi.org/10.7554/eLife.32847.010

The following source data and figure supplement are available for figure 3:

**Source data 1.** Quantification of OPCs and EPCs.
DOI: https://doi.org/10.7554/eLife.32847.012

**Figure supplement 1.** Extended analysis of pericardial markers in EGF loss- and gain-of-function backgrounds.
DOI: https://doi.org/10.7554/eLife.32847.011

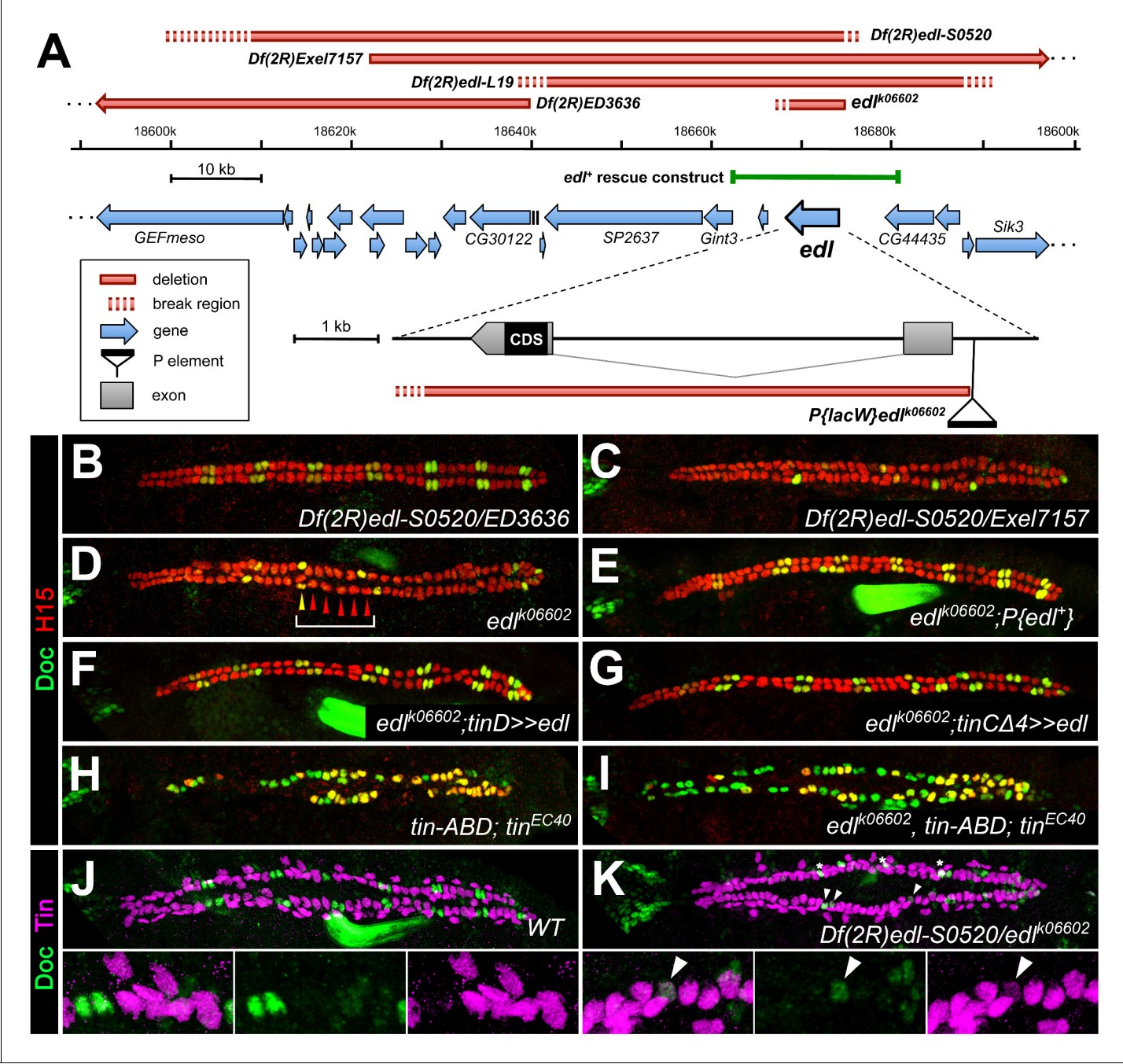

**Figure 4.** Edl is a decisive factor of ostial cardioblast specification. (**A**) Map of the *edl* locus with the used alleles and deficiencies. (**B–I**) Doc2+3/H15 stainings as in *Figure 1*. (**B**) Embryo with transheterozygous combination of *Df(2R)edl-S0520* (*edl* deleted) and *Df(2R)ED3636* (*edl* present) showing a regular '2 + 4' CB pattern of oCBs and gCBs. By contrast, amorphic *edl* mutants *Df(2R)edl-S0520/Exel7157* (**C**) and *edl^{k06602}* (**D**) have only few oCBs. Note the occurrence of '1 + 5' CB patterns (bracket). (**E**) The regular CB pattern is restored by a genomic *edl^+* transgene. A nearly normal CB pattern is observed in *edl* mutants upon expression of *UAS-edl* in the dorsal mesoderm via *tinD-GAL4* (**F**) or only in CBs or their progenitors via *tinCΔ4-GAL4* (**G**). In cardioblast-specific *tin* mutants (carrying a rescue construct for early *tin* function) all CBs present become Doc⁺, irrespective of whether *edl* is functional (**H**) or not (**I**). Observation of some H15⁻ Doc⁺ CBs in (**H**) and (**I**) suggest that robust H15 expression requires normal *tin* function. (**J**) Mutually exclusive expression of Doc and Tin proteins in the wild type at late stage 15. (**K**) In *edl* mutants, Doc and Tin are co-expressed in some CBs (arrowheads). These oCBs display either low level expression of both Tin and Doc (as exemplified in the magnification) or low levels of Tin concurrent with close to normal levels of Doc. Asterisks denote positions of artificial signal overlap due to co-projection of oCBs and TPCs.

DOI: https://doi.org/10.7554/eLife.32847.013

The following figure supplements are available for figure 4:

*Figure 4 continued on next page*

*Figure 4 continued*

**Figure supplement 1.** Cardiac *edl* expression.
DOI: https://doi.org/10.7554/eLife.32847.014
**Figure supplement 2.** Dynamic expression of *edl* in the cardiogenic mesoderm as detected by an intron-specific probe.
DOI: https://doi.org/10.7554/eLife.32847.015

ratio (*Figure 5I*). Thus, we suggest that Edl acts mainly via negative modulation of PntP2 activity during cardioblast diversification.

An additional function of Pnt (and thereby Edl) regarding to the total number of CBs is also apparent in *Figure 5*. The increase in the total CB number detected in *pnt* mutants is reminiscent of Notch pathway mutants. *Figure 5—figure supplement 1* shows examples of such mutants isolated from our EMS screen. There is an important difference between *pnt* and Notch pathway mutants regarding the oCBs:gCBs ratio. Whereas oCBs account for about 40–50% of the CBs in *pnt* mutants (as compared to 27% in the wild type), all Notch pathway mutants for which CB patterning data are available feature a significantly smaller fraction of oCBs than *pnt* mutants (*Figure 5—figure supplement 1D*). The maximum fraction of oCBs observed was 33% of the total CB number, found in *mam$^{S0669}$*. In *kuz* mutants (data not shown; *Albrecht et al., 2006*), oCBs even increase by smaller factors than gCBs resulting in oCB fractions below 27%. (Some differences in the oCBs:gCBs ratio between various Notch pathway mutants are likely to arise from variable impact on lateral inhibition and specific functions of Notch in asymmetrically dividing lineages). On a side note, *edl* expression, which was found to be positively regulated by Notch signaling in a *Drosophila* cell culture system (*Krejcí and Bray, 2007*), is not negatively affected in the cardiogenic mesoderm of two *mam* alleles and in *bib$^{S1538}$* mutants (*Figure 5—figure supplement 2* and data not shown).

## Edl and Pnt regulate ostial fate by controlling *seven-up* expression

The population of oCBs is characterized by expression of *svp*. In *svp* mutants all oCBs are converted into Tin$^{+}$/Doc$^{-}$ CBs due to de-repression of *tin* (*Gajewski et al., 2000*; *Lo and Frasch, 2001*; *Zaffran et al., 2006*; *Figure 6—figure supplement 1A*). Therefore, we tested the possibility that Edl promotes oCB fate by regulating *svp*. In the wild type, expression of *svp* is recapitulated by the enhancer trap *svp$^{AE127}$-lacZ* (*Figure 6A*; *Lo and Frasch, 2001*). In *edl* mutants, *svp*-LacZ expression is strongly reduced in cardiac cells (*Figure 6B,D*). The reduction in numbers of both *svp*-LacZ$^{+}$ oCBs and OPCs at late stages (*Figure 6D* cf. 6C) suggests that *edl* already affects the fates of their common progenitors. Consistent with a function in promoting *svp* expression and oCBs fates, mesodermal overexpression of *edl* leads to larger numbers of *svp*-LacZ$^{+}$ cardiac cells, particularly of CBs, where *svp* expression correlates with expanded Doc expression (*Figure 6E,F*). As shown for *Doc* expression, *svp* expression can be suppressed by PntP2 hyperactivity (green asterisks in *Figure 6H*). These observations and further evaluation of the epistatic relations between *svp* and *edl* (*Figure 6—figure supplement 1*) demonstrate that *edl* affects CB patterning by blocking Pnt activity upstream of *svp*.

## Cardioblast subtype-specific expression of the PntP1 isoform is regulated by PntP2 and Edl

Proposing a gCB-specific function of Pnt, we next analyzed its cardiac expression. Boisclair Lachance et al. previously reported that the expression of a fully functional genomic *pnt-GFP* transgene mirrors the combined expression of all Pnt isoforms (*Boisclair Lachance et al., 2014*). The authors detected Pnt-GFP fusion protein in nearly all cells of the cardiac region, but highest levels were observed in two Yan-negative clusters per hemisegment flanking Eve$^{+}$ cells. We confirmed and refined these observations showing that high levels of Pnt-GFP are present in the nuclei of gCB progenitors as identified by their position, characteristically enlarged size, presence of only low levels of Doc, and absence of *svp*-LacZ expression (*Figure 7A*). We attribute these high total Pnt levels largely to a gCB-specific expression of the PntP1 isoform since PntP1-specific antibodies (*Alvarez et al., 2003*) specifically label gCB progenitors (*Figure 7B*), whereas *pntP2* transcripts are present in a rather uniform pattern in the mesoderm including the cardiogenic area (*Klämbt, 1993*; and data not shown). We further speculated that PntP2 could activate *pntP1* transcription in gCB progenitors for a sustained signaling response as found in other tissues (*Shwartz et al., 2013*). This assumption is indeed supported by our genetic data. First, we detect PntP1

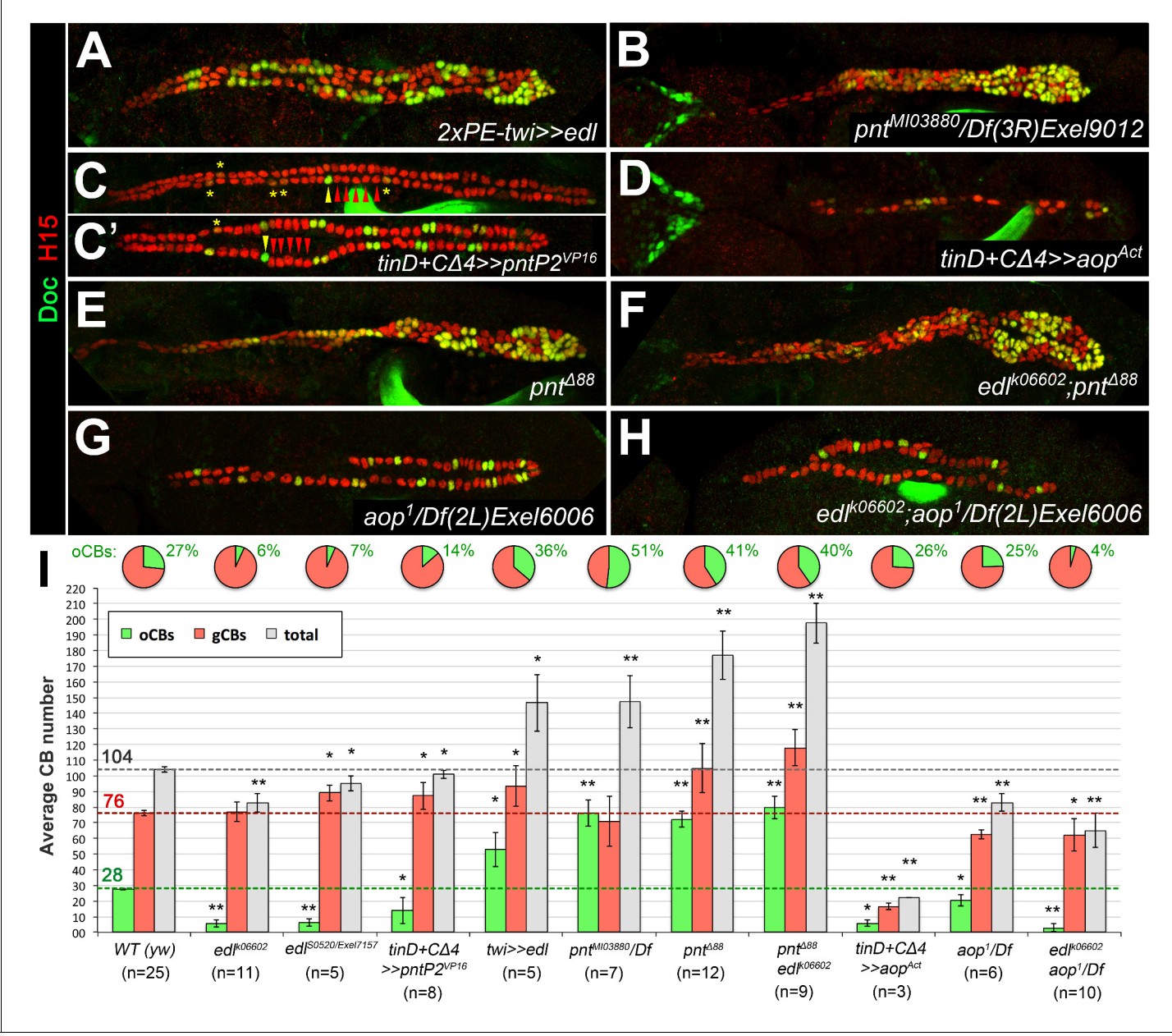

**Figure 5.** Edl promotes oCB fate via inhibition of PntP2. (A–H) CB pattern in embryos with modified activity of *edl* and/or genes encoding the ETS proteins Pnt and Yan revealed by H15/Doc2+3 stainings. (A) Pan-mesodermal *edl* overexpression via *twist-GAL4* leads to extra CBs with a disproportionately high increase in oCB numbers. This phenotype is reminiscent to that of the *pnt* mutants *pnt$^{MI03880}$* (a PntP2-specific mutant; here in trans with a *pnt*-deleting deficiency, (B) and *pnt$^{Δ88}$* (without any functional Pnt isoform, (E). (C,C') Conversely, an *edl* mutant-like phenotype (loss/ conversion of oCBs, exemplified by arrowheads for one hemisegment, and CBs with low Doc levels marked by asterisks) is generated by overexpression of a constitutively active PntP2 variant in the dorsal/cardiogenic mesoderm. C and C' depict strong and weak phenotypes, respectively. (D) Overexpression of the constitutively active repressor Yan/Aop leads to a loss of both gCBs and oCBs. (E,F) The CB phenotypes of *pnt* and *edl pnt* double mutants are very similar suggesting that *edl* acts mainly by blocking Pnt activity during CB specification. (G) Hemizygous *aop* mutant showing a moderate reduction of both CB types. (H) *edl aop* double mutant combining *aop*-like and *edl*-like defects. (I) Quantification of cardioblasts in various genotypes affecting Edl, Pnt or Yan/Aop activities (annotated as in *Figure 1M*).

DOI: https://doi.org/10.7554/eLife.32847.016

The following source data and figure supplements are available for figure 5:

**Source data 1.** Quantification of cardioblasts in various genotypes affecting Edl, Pnt or Yan/Aop activities.

DOI: https://doi.org/10.7554/eLife.32847.020

**Figure supplement 1.** The numbers of both generic and ostial cardioblasts increase upon mutation of genes involved in Notch signaling.

*Figure 5 continued on next page*

Figure 5 continued

DOI: https://doi.org/10.7554/eLife.32847.017

**Figure supplement 1—source data 1.** Quantification of cardioblasts in Notch signaling-related genotypes.

DOI: https://doi.org/10.7554/eLife.32847.018

**Figure supplement 2.** Expression of *edl* in the cardiogenic mesoderm is still observed in Notch signaling-related mutants.

DOI: https://doi.org/10.7554/eLife.32847.019

in an expanded pattern in the cardiogenic mesoderm of *edl* mutants in which PntP2 activity is assumed to increase (*Figure 7C*). Second, overexpression of *edl* (i.e. repression of PntP2 function) as well as genetic disruption of *pntP2* resulted in a near-complete loss of cardiac PntP1 (*Figure 7D,E*; note persistent expression of PntP1 in other cells located more laterally). We conclude that the combined activities of Edl and PntP2 lead to the confined *pntP1* expression in gCBs. The EGF Spitz appears to be a major, although not necessarily the sole factor for the MAPK-mediated activation of PntP2 in this context, because PntP1 levels are reduced but not eradicated in cardiac cells of amorphic *spi* mutants (*Figure 7F*).

## The Tbx20 ortholog Midline contributes to Pnt-dependent repression of *svp* in the working myocardial lineage

According to the common view, we expect Pnt to act as a transcriptional activator also during CB diversification, particularly since overexpression of PntP2 fused to the VP16 activator domain has essentially the same effect on cardiac patterning as PntP1 overexpression (*Figure 6H* and data not shown). Therefore, its negative impact on *svp* expression is likely to involve Pnt-dependent activation of a transcriptional repressor. Interestingly, the T-box factor Midline (Mid), like PntP1, shows expression in early gCB progenitors (*Figure 2I,K*; *Figure 7G*). We previously reported that *mid* functions to maintain *tin* expression in gCBs, thereby restricting *Doc* expression to oCBs (*Reim et al., 2005*). Consistent with this function our EMS screen also generated novel *mid* alleles showing the same CB patterning defects as previously described alleles (*Supplementary file 1*-Table S1, *Figure 7H* and data not shown). While a direct regulation of *tin* by Mid was previously proposed to be responsible for these changes (supported by the gain- and loss-of-function phenotypes of *mid*; *Qian et al., 2005*; *Reim et al., 2005*), another non-exclusive scenario could involve repression of *svp* (encoding a repressor of *tin*) by Mid. Consistent with the latter, we observe a *Doc*-like expansion of *svp* expression in *mid* loss-of-function mutants (*Figure 7I*) and a reduction of *svp* expression upon ectopic overexpression of *mid* via *tinD +tinCΔ4-GAL4* (*Figure 7J*). Moreover, persistent *tin* expression in all CBs of *mid svp* double mutants (*Figure 7—figure supplement 1D*, compare to control in A and single mutants in B and C) demonstrates that *mid* is not directly required for *tin* expression in CBs. Furthermore, the wild type-like expression of *svp-lacZ* (with nearly no LacZ in gCBs) observed in the same genetic background argues for the involvement of a Svp-dependent positive feedback loop in ectopic cardiac *svp* activation in gCBs, as has been predicted previously based on *svp* overexpression studies (*Zaffran et al., 2006*). The cardiac pattern phenotype of *edl mid* double mutants is a composite of the single mutant phenotypes. The number of oCBs (average oCBs: 24.4 ± 3.6; n = 6) is strongly increased as compared to *edl* mutants, but reduced in comparison with *mid* mutants, with total CB numbers being similar to those of *edl* mutants. In some cases, a near wild-type pattern is observed (*Figure 7K*), although many embryos display an asymmetric arrangement of CBs. While the prevalence of many Doc-negative CBs in this background implies that *mid* is not the only factor that limits oCB fate, it also indicates that *edl* is normally required in the oCB lineage to restrict *mid* activity, possibly by blocking a Pnt-dependent activation of *mid* transcription. This hypothesis is indeed supported by the reversion of ectopic *Doc* and *svp* expression in *pnt* mutants upon forced *mid* expression (*Figure 7L*, *Figure 7—figure supplement 2C*). By contrast, overexpression of the previously assumed Mid target *tin* in this background only represses *Doc*, but not *svp* (*Figure 7—figure supplement 2D*).

To further test the idea that Mid is a repressor of oCB fate downstream of *pnt*, we analyzed whether it is a direct target of Pnt. Notably, an enhancer identified as a Tin target and named *midE19* (*mid180* for a shorter minimal version) was recently shown to drive *mid* expression specifically in gCBs (*Jin et al., 2013*; *Ryu et al., 2011*; *Figure 8A–C*; *Figure 9A,C*). Since this enhancer does not drive reporter expression in oCBs after germ band retraction as detected for *mid* in the genomic context, additional *cis*-regulatory regions must be at work to reproduce all aspects of cardiac *mid* expression. The characteristic activity pattern of the enhancer suggests that this regulatory region may be specifically (or exclusively) devoted

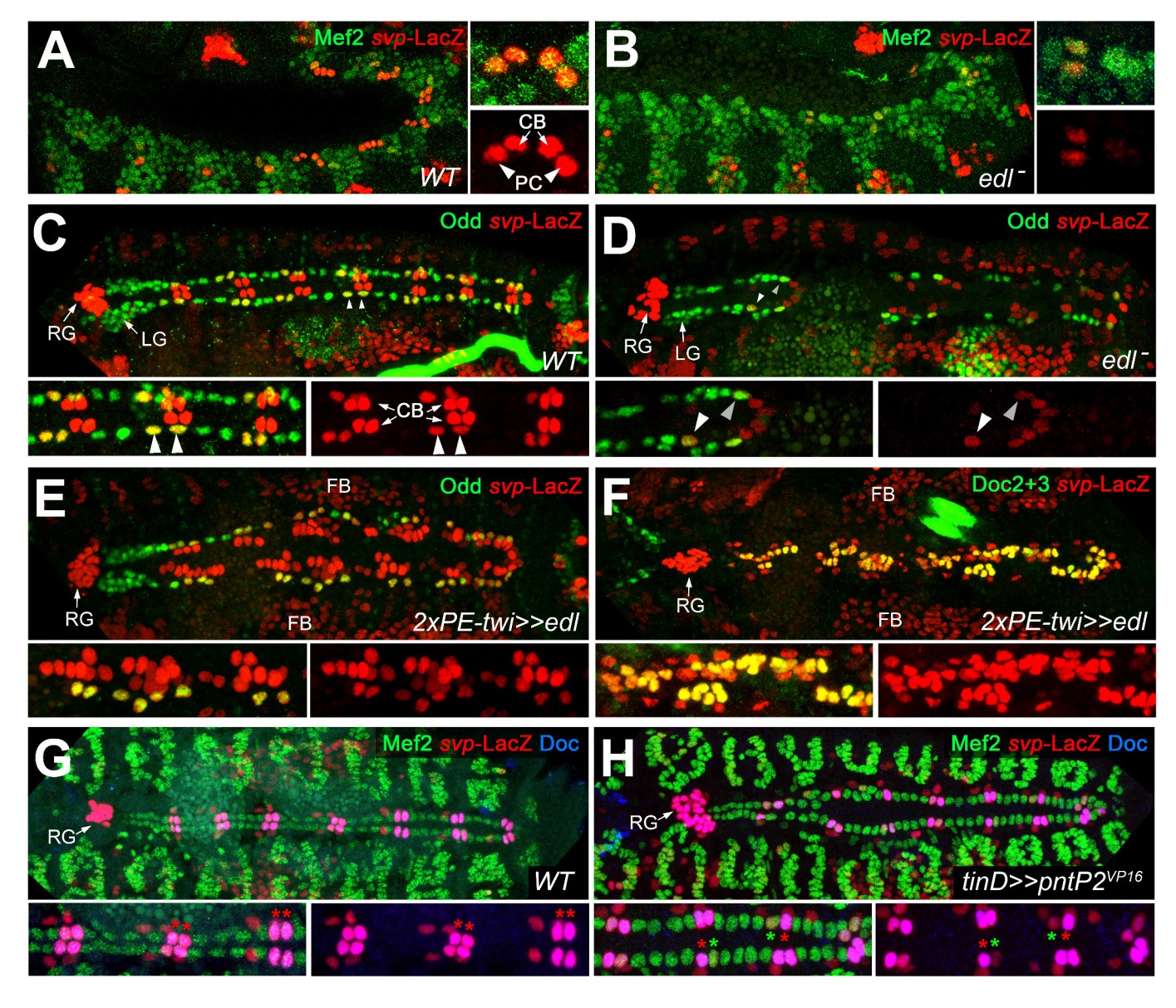

**Figure 6.** Edl is required for *svp* expression. (**A**) In stage 12 control embryos (lateral view) carrying one copy of *svp*$^{AE127}$-*lacZ*, β-galactosidase is detected in oCBs (arrows) and their sibling OPCs (arrowheads) within the Mef2-labeled mesoderm. (**B**) Cardiac *svp*-LacZ expression is strongly reduced in *edl* mutants (*Df(2R)edl-S0520/Exel7157;svp*$^{AE127}$-*lacZ/+*). (**C–E**) Odd/*svp*-LacZ staining in stage 16 embryos. (**C**) In the control, each hemisegment contains two oCB-related *svp*-LacZ$^+$ OPCs and two *svp*-LacZ$^-$ OPCs. The total number of OPCs decreases if *edl* is absent (*Df(2R)edl-S0520/edl-L19; svp*$^{AE127}$-*lacZ/+*) (**D**) or overexpressed (**E**), but different OPC subpopulations account for these losses: *svp*-LacZ$^+$ OPCs (arrowheads) are reduced in *edl* mutants, *svp*-LacZ$^-$ OPCs in *edl* overexpressing embryos. (**E,F**) Pan-mesodermal overexpression of *edl* leads to a drastic increase in the number of *svp*-LacZ$^+$/Doc$^+$ cardioblasts (Odd$^-$). Compare F to the control in *Figure 3F*. (**G,H**) Mef2/Doc2+3/β-galactosidase staining in *svp-lacZ/+* controls (**G**) and embryos overexpressing constitutively active *pntP2*$^{VP16}$ in the dorsal mesoderm (**H**). Overexpression of *pntP2*$^{VP16}$ via *tinD-GAL4* leads to significantly reduced *svp* and *Doc* expression (examples labeled with green asterisks; average number of Svp$^+$ CBs: 20.6 ± 3.0, p=0,00069**; accompanied by an increased number of Svp$^-$ CBs: 83.4 ± 2.6, p=0.00015**; n = 7) as compared to normal oCBs (red asterisks). LG: lymph gland, RG: ring gland, FB: fat body.

DOI: https://doi.org/10.7554/eLife.32847.021

The following figure supplement is available for figure 6:

**Figure supplement 1.** Epistatic relationship between *edl* and *svp*.
DOI: https://doi.org/10.7554/eLife.32847.022

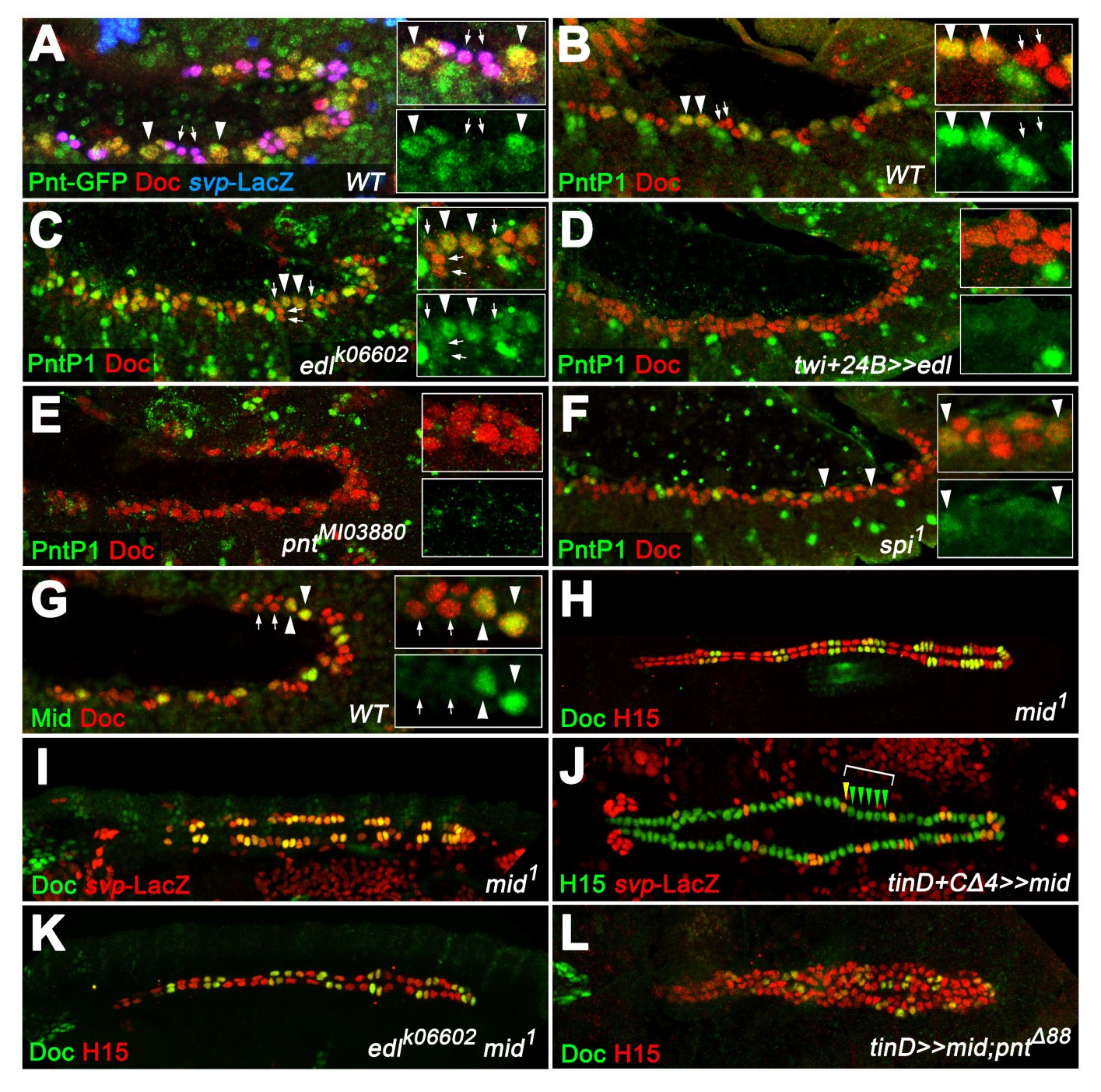

**Figure 7.** PntP1 and Mid are specifically expressed in early gCB progenitors to antagonize oCB fate. (A) Detection of Doc3+2, β-galactosidase and GFP-tagged Pnt (all isoforms) in a *pnt-GFP/+; svp^{AE127}-lacZ/+* embryo at the beginning of stage 12 (lateral view). Highest levels are observed in gCB progenitors (large *svp*-LacZ-negative nuclei with low levels of Doc, arrowheads) and low levels in oCBs and their siblings (small *svp*-LacZ+ nuclei with higher Doc levels, arrows). (B) At the onset of germ band retraction, PntP1 becomes expressed in gCB progenitors (arrowheads) of wild type embryos. Cardiac cells are labeled via anti-Doc3+2 staining. PntP1 is not detected in oCBs and their siblings (arrows). (C) In *edl*- mutants cardiac PntP1 expression is generally increased and detected ectopically in some small nuclei that correspond to prospective oCBs and their siblings (arrows). (D) Pan-mesodermal overexpression of *edl* leads to a strong decrease of cardiac PntP1 expression while other mesodermal tissues are less affected. (E) The same effect is seen in *pntP2* mutants. (F) In *spi* mutants PntP1 levels are reduced as well, although not as severely as upon loss of *pntP2* function. (G) Like PntP1, Mid protein is found in gCB progenitors (arrowheads), but not in prospective oCBs (arrows) at the beginning of germ band retraction. (H,I) The cardiac phenotype of *mid* mutants is characterized by variable expansion of Doc, which largely correlates with ectopic *svp* expression in CBs (I,

*Figure 7 continued on next page*

*Figure 7 continued*

normal pattern shown in *Figure 3F*). (J) Overexpression of *mid* represses *svp* expression in H15-labeled cardioblasts (arrowheads indicate a hemisegment with five lacZ-negative nuclei). (K) Combining homozygous *mid* and *edl* mutations results in the restoration of oCBs in comparison to *edl* single mutants (*Figure 4D*), suggesting that *edl* normally antagonizes *mid* function. An additional *edl* function regarding the total CB number is not rescued by abrogation of *mid*. (L) Overexpression of *mid* in the dorsal mesoderm via *tinD-GAL4* in a *pnt* null background converts many of the extra oCBs into gCBs (cf. *Figure 5E*).

DOI: https://doi.org/10.7554/eLife.32847.023

The following figure supplements are available for figure 7:

**Figure supplement 1.** Expression of *tin* in gCBs is indirectly stabilized by *mid* via *svp* repression.

DOI: https://doi.org/10.7554/eLife.32847.024

**Figure supplement 2.** Additional data supporting *mid* function in gCBs.

DOI: https://doi.org/10.7554/eLife.32847.025

to the reception of early gCB-specific inputs. Consistent with our assumption that this enhancer is also a target of Pnt, very little *midE19*-GFP activity is detectable in *pnt* mutants (*Figure 8D*), reduced activity is observed in embryos with mesodermal *edl* overexpression (*Figure 8E*), and expanded activity is seen upon overexpression of PntP1 (*Figure 8F*; note occasional expansion into CBs with no detectable Tin) or PntP2$^{VP16}$ (not shown). An observed reduction of *midE19*-driven GFP levels in many of the retained Tin$^+$ gCBs of *rho* mutants (*Figure 8G*) corroborates that EGF signaling feeds into *mid* activation. The idea that *mid* is a target of Pnt is further supported by the almost complete elimination of reporter activity upon mutating a single ETS binding motif within the mid180 minimal cardiac enhancer (*Figure 9A–D*) as well as the strong reduction of endogenous *mid* transcription in emerging CBs during germ band retraction stages in *pnt* mutants (*Figure 9E–H*). After germ band retraction, endogenous *mid* is activated independently of *pnt* in all CBs (*Figure 9J*) as observed in the wild type (*Figure 9I*) indicating that distinct mechanisms regulate *mid* transcription in early gCB progenitors and maturing CBs.

In sum, our data lead to the conclusion that EGF signaling contributes to gCB specification by at least two distinct mechanisms, Pnt-independent specification of a subset of cardiac progenitors as well as Pnt-dependent inhibition of ostial cardioblast fate. Modulation by Edl is needed to inhibit Pnt-dependent gene activation and thus enable formation of ostial cardioblasts.

## Discussion

The specification and diversification of particular cell types are linked to the establishment of lineage-specific transcriptional programs. The differences in these programs are often prompted by distinct local signaling activities. The cells in the early heart fields of *Drosophila* acquire their cardiogenic potential by intersecting BMP and Wnt signal activities (*Frasch, 1995*; *Reim and Frasch, 2005*; *Wu et al., 1995*), but cell diversification within this area requires additional regulatory inputs. Previous studies established that progenitors of cardioblasts, pericardial cells and dorsal somatic muscles are selected by RTK/Ras/MAPK signaling, whereas lateral inhibition by Delta/Notch signaling activity counteracts this selection in neighboring non-progenitor cells (*Carmena et al., 2002*; *Grigorian et al., 2011*; *Hartenstein et al., 1992*). The progenitors of the definitive cardiogenic mesoderm, which give rise to all cardiac cells except for the somatic muscle lineage-related EPCs, co-express the cardiogenic factors Tin, Doc and Pnr, a unique feature that separates them from other cells (*Reim and Frasch, 2005*). In addition to limiting the number of progenitors, Notch signaling has a second function during *Drosophila* cardiogenesis that promotes pericardial (or in thoracic segments, hematopoietic) over myocardial fate (*Albrecht et al., 2006*; *Grigorian et al., 2011*; *Hartenstein et al., 1992*; *Mandal et al., 2004*). Other factors previously reported to impose heterogeneity in the heart field include the cross-repressive activities of the homeodomain factors Eve and Lbe (*Jagla et al., 2002*) as well as ectoderm-derived Hedgehog (Hh) signals (*Liu et al., 2006*; *Ponzielli et al., 2002*). In segmental subsets of cardioblasts, Hh signaling was proposed to act as a potential activator of *svp* in prospective oCBs (*Ponzielli et al., 2002*) but whether these are direct or indirect effects of Hh on these cells has not been ascertained.

Based on the findings of our study, we present a novel model of cardioblast diversification that introduces EGF signaling activities and lineage-specific modulation of the MAPK effector Pointed by Edl as crucial factors for the specification of generic working myocardial and ostial cell fates (*Figure 10*). We

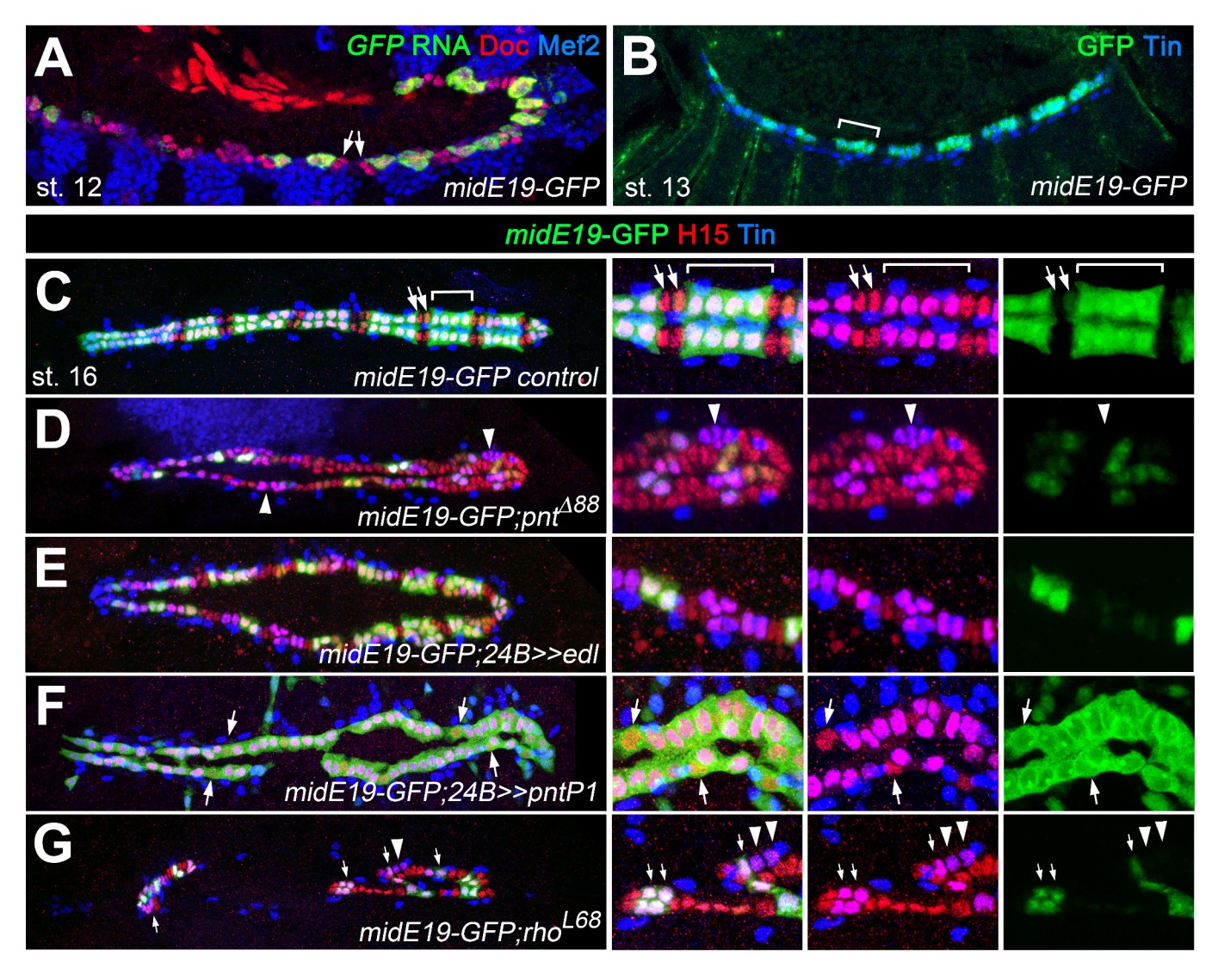

**Figure 8.** Characterization of a Pnt-responsive *mid* enhancer. (A–C) Expression analysis of the *midE19-GFP* reporter in the wild type background showing segmental expression in gCB progenitors at stage 12 (A: co-expression of *GFP* RNA, Mef2 and low levels of Doc) and later in the Tin⁺/H15⁺ gCBs (bracket; B: stage 14 stained for GFP protein and Tin; C: stage 16 stained for GFP, Tin and H15 proteins). No or very little reporter expression is detectable in oCBs and their presumed precursors (arrows). (D) Despite an overall increase in CB number, *midE19*-GFP expression is severely reduced in amorphic *pnt* mutants. Most of the Tin⁺/H15⁺ gCBs (purple nuclei, arrowheads) lack GFP expression. (E) Mesodermal overexpression of *edl* via *how^{24B}*-GAL4 also leads to a loss of *midE19*-GFP in many gCBs. (F) Overexpression of *pntP1* via *how^{24B}*-GAL4 leads to nearly continuous *midE19*-GFP expression in CBs. In some instances, the reporter is activated even in Tin⁻ CBs (arrows). (G) Loss of *rho* function, which is expected to cause reduced PntP2 activity, leads to a complete loss of GFP in some of the retained gCBs (arrowheads) and a level reduction in others (arrows). In comparison to *pnt* mutants (D), a higher fraction of gCBs retains substantial GFP expression indicating additional, *rho*-independent inputs upstream of Pnt.

DOI: https://doi.org/10.7554/eLife.32847.026

propose that EGF/MAPK signaling promotes the development of generic working myocardial progenitors (red cell in *Figure 10*) by two mechanisms that differ in their requirement for the ETS protein Pnt:

1. EGF promotes the correct selection and specification of gCB progenitors. This is evident from our loss- and gain-of-function analysis of EGF signaling components. This EGF function is obviously independent of *pnt*, since *pnt* null mutants display excessive numbers of CBs (with gCB numbers comparable to the wild type or even increased), a phenotype different from that of

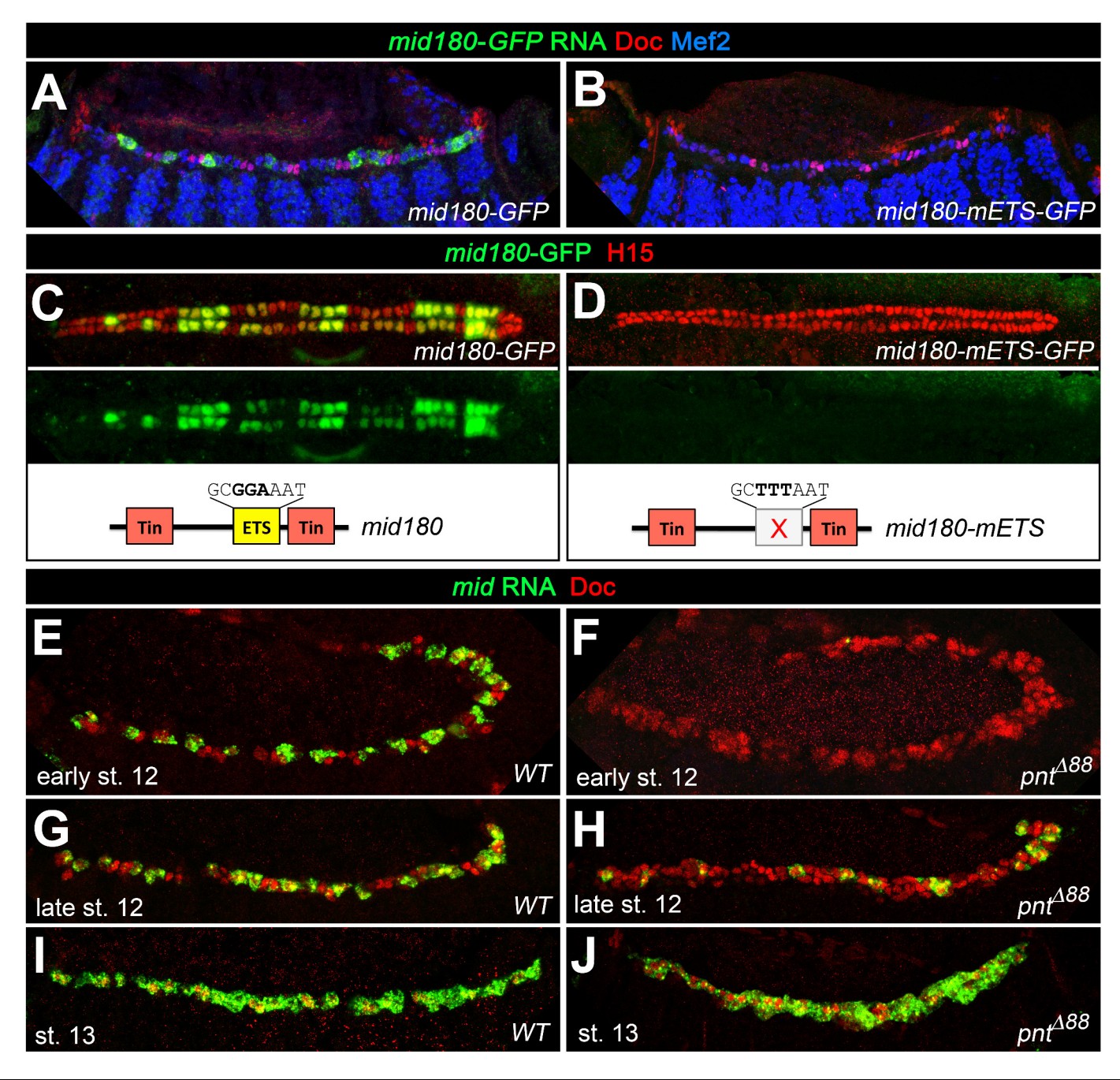

**Figure 9.** Additional experimental support for the regulation of *mid* by the ETS factor Pnt. Expression of GFP RNA (A, stage 13) and protein (C, stage 16) driven by the minimal cardiac *mid* enhancer, *mid180*, is less robust than *midE19*-GFP but shows essentially the same expression pattern. The minimal enhancer contains a single ETS binding motif flanked by two Tin-binding sites (indicated in the scheme below). (B,D) Mutating the ETS-binding site leads to near-complete abolishment of *mid180*-GFP expression. (E–J) Analysis of *mid* mRNA expression in cardiac cells doubly stained with anti-Doc3+2 antibody. In the wild type, *mid* mRNA is first detected in gCB progenitors at early stage 12 (E); its expression begins to expand during germ band retraction (G) until it reaches continuous expression in all CBs at stage 13 (I). By contrast, amorphic *pnt* mutants show reduced cardiac *mid* expression during germ band retraction (F,H). Regular uniform *mid* expression is observed only after germ band retraction (J).

DOI: https://doi.org/10.7554/eLife.32847.027

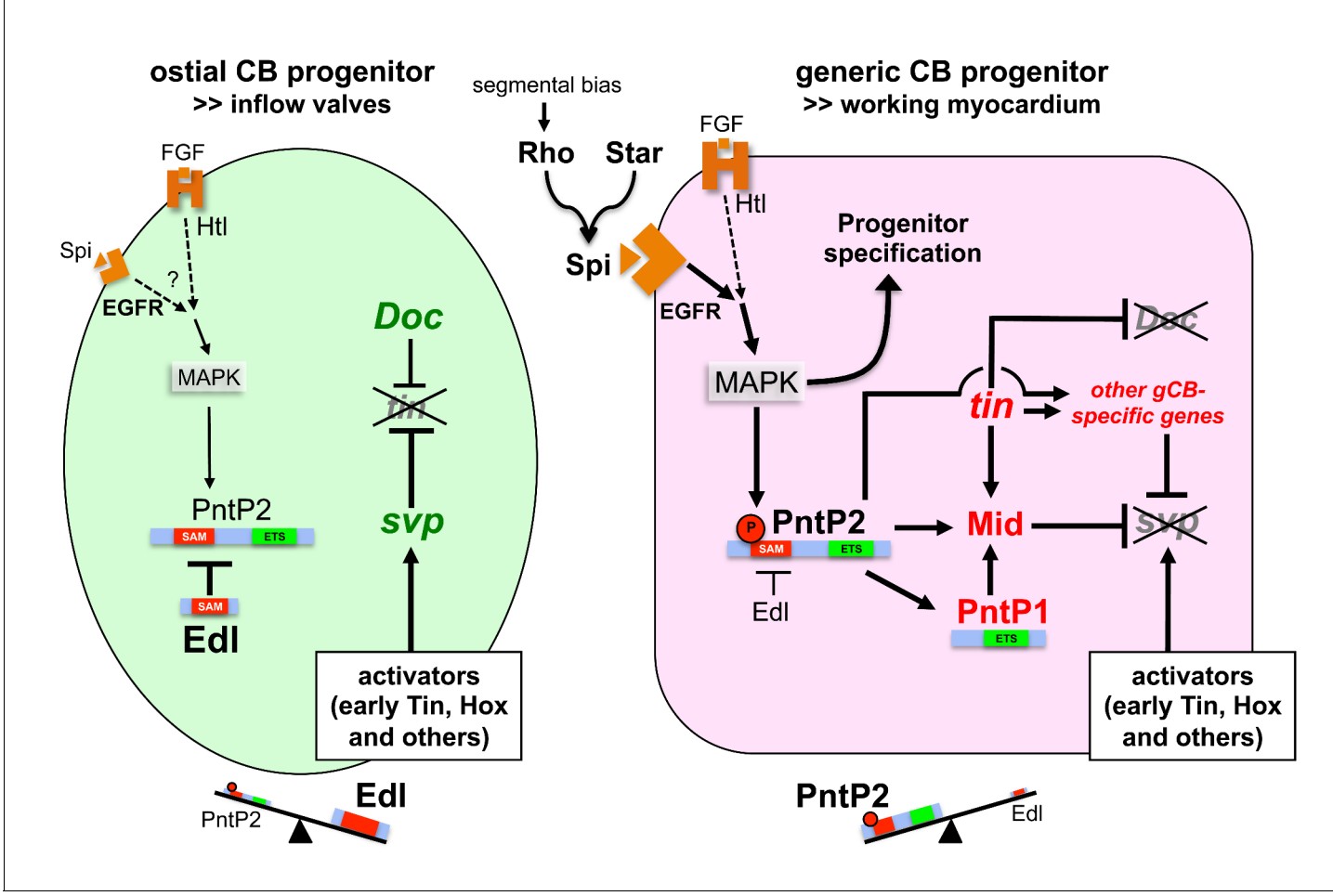

**Figure 10.** Model of regulatory interactions in generic and ostial CB progenitors. Genes activated in a subtype-specific manner in gCB or oCB progenitors are colored in red and green, respectively. Larger font sizes and thicker lines indicate higher levels. Dashed lines indicate presumed regulations. In principle, MAPK can be activated in cardiac progenitors by EGF/EGFR and FGF/Htl signals. Generic cardioblast development depends on EGF-activated MAPK signaling which provides *pnt*-independent and *pnt*-dependent functions. The suppression of *svp* and subsequent regulation of *tin* and *Doc* is a *pnt*-dependent function that is in part mediated by activation of *mid* in presumptive gCBs. This step is likely to be supported by the gCB-specific expression of constitutive active PntP1. The gCB-specific cascade may require a higher level of MAPK activity to overcome the blockage of PntP2 by Edl. Alternatively or in addition, Edl levels might be differentially regulated in gCBs and oCBs by yet unknown mechanisms. In oCB progenitors, Edl keeps activated PntP2 below a critical threshold leading to absence or delayed onset of expression of oCB fate antagonists such as *mid*. This in turn permits *svp* activation by Hox genes and Tin derived from early stages. Presumed transcriptional activators of *svp* acting downstream of segmental Hh signals in oCB progenitors are not mandatory in this model, although it does not categorically exclude such contributions. Some details and additional interactions have been omitted for clarity. For a more complex version of the model see the corresponding figure supplement.

DOI: https://doi.org/10.7554/eLife.32847.028

The following figure supplement is available for figure 10:

**Figure supplement 1.** Extended model of regulatory interactions in generic and ostial CB progenitors.

DOI: https://doi.org/10.7554/eLife.32847.029

mutants defective in EGF pathway components upstream of Pnt (*Alvarez et al., 2003*); this study).

2. EGF signals affect the diversification of CB progenitors by impinging on a PntP2-dependent transcriptional cascade that eventually leads to suppression of $Tin^-$ oCB and the adoption of $Tin^+$ gCB fates. This function is mediated by stimulating the gCB progenitor-specific expression of regulatory genes such as *mid* (depicted in red in *Figure 10*), which in turn will promote transcription of gCB-specific differentiation genes and/or repression of oCB-specific factors (depicted in green in *Figure 10*).

Since this study focuses mainly on the second, Pnt-dependent cardioblast diversification function, we elucidate the regulatory circuitry within each cardioblast lineage more extensively in the paragraphs further below. Prior to that, we briefly discuss our findings regarding the EGF signaling function during CB progenitor formation.

## EGF signaling and cardiac progenitor selection

According to our data, EGF signals are the major source for MAPK activation and progenitor specification in the symmetrically dividing progenitors of gCBs and OPCs (and likely also TCPs). By contrast, EGF signals are dispensable (in high doses even unfavorable) for the development of progenitors of oCBs and their sibling OPCs. Thus, EGF signaling clearly has a lineage-specific function, which is most easily explained by a requirement for progenitor selection and cell fate specification. This interpretation does not preclude contributions to cell survival (which might depend on differentiation) or lineage-specific divisions (i.e. correct progenitor specification is a prerequisite of the subsequent final division). Notably, in most hemisegments of the analyzed EGF pathway mutants, the number of gCBs is reduced by even numbers and remaining gCB pairs are usually of the same subtype regarding Lbe expression, arguing for a requirement prior to completion of the final mitotic division at the progenitor stage. Since we have only minor evidence for apoptosis and fate conversions into other cell types in EGF-related mutants (minor increase in oCBs, overall reduction of PCs) we propose that many of the missing gCBs are not selected as highly Delta-expressing CB progenitors upon reduced MAPK signaling activity (*Carmena et al., 2002*; *Grigorian et al., 2011*; *Hartenstein et al., 1992*). Instead, they are likely retained by default within a pool of undifferentiated dorsal mesoderm cells.

Our overexpression studies demonstrate that the timing of EGF signals is crucial for their function in differential progenitor specification. In previous studies, earlier functions of MAPK signaling might have obscured its specific impact on gCBs and OPC subtypes. While early pan-mesodermal activation of MAPK signaling or expression of constitutive active Pnt forms via the *twi-GAL4* driver reduces the numbers of all cardiac cells except the Eve$^+$ progenitors (*Alvarez et al., 2003*; *Bidet et al., 2003*; *Liu et al., 2006*; and our own data), later MAPK activation favors formation of the symmetrically dividing OPC, TPC and gCB progenitor subpopulations (e.g. as seen in our experiments with *tinD-GAL4*-driven *rho*). We propose that the specification of these progenitors requires the context of the definitive cardiogenic mesoderm, whereas premature MAPK activation in all mesoderm cells negates any pro-cardiogenic effects due to the massive expansion of Eve$^+$ clusters (which are normally the first cells in the heart field to display MAPK and *rho* activity) at the expense of the cardiac progenitors in the neighboring C14/C16 clusters (*Buff et al., 1998*; *Jagla et al., 2002*; *Liu et al., 2006*; *Qian et al., 2005*; and our own data not shown).

As discussed above, cardioblast formation as such is independent of *pnt*. How could this be achieved? Growth factor-activated MAPK can also phosphorylate the repressor Yan thereby diminishing its activity as an antagonist of progenitor selection (*Halfon et al., 2000*; *O'Neill et al., 1994*; *Rebay and Rubin, 1995*). Therefore, it is conceivable that MAPK activity in the context of CB progenitor selection might be primarily required to eliminate the repressive activity of Yan. This would be consistent with the observed reduction of cardiac cells upon *aop/yan* hyperactivation (*Halfon et al., 2000*; this study). In this context, a minor function of Edl could contribute to the robustness of cardiac progenitor selection and thus total cardioblast and pericardial cell numbers by reducing the repressive Yan activity.

## A novel model for cardioblast diversification connecting EGF signaling, ETS protein activity and lineage-specific transcription factor patterns

Combining previous findings with our new data we have conceived the regulatory model of cardioblast diversification illustrated in *Figure 10*. The central element of this model is the differential modulation of Pnt activity in the gCB and oCB progenitors leading to lineage-specific outcomes.

### Basic features of gene regulation in the gCB lineage

We identified *mid* as a key target gene of Pnt in gCB progenitors based on its early gCB-specific expression, Pnt-dependent transcriptional regulation and its ability to repress the oCB-specific regulator gene *svp*. Since Svp represses *tin* expression (*Gajewski et al., 2000*; *Lo and Frasch, 2001*), *svp* suppression

provides an important part of the explanation for the previously reported positive role of Mid in maintaining *tin* expression in gCBs (*Qian et al., 2005*; *Reim et al., 2005*). Furthermore, expanded expression of *tin* in *mid svp* double mutants argues against the possibility that Mid stimulates *tin* expression directly. While Tin acts as a repressor of *Doc* via unknown mechanisms in gCBs, it does not repress *svp* (*Zaffran et al., 2006*; *Figure 7—figure supplement 2D*). On the contrary, at least in the early cardiogenic mesoderm, it acts as an activator of *svp* in oCB progenitors (*Ryan et al., 2007*). Thus, in the absence of appropriate repressors such as Mid, *svp* expression can expand into gCBs.

## Basic features of gene regulation in the oCB lineage

In prospective oCB progenitors, Pnt activity must be kept in check to permit *svp* expression and thereby *tin* repression and *Doc* activity. Fittingly, we identified *edl*, a gene linked to negative regulation of MAPK signaling and cell identity determination in several tissues - including the eye (*Yamada et al., 2003*) and recently in certain somatic muscle progenitors (*Dubois et al., 2016*) - as a novel regulator in the context of cardiac cell specification, particularly that of oCB progenitor fate (green cell in *Figure 10*). This function is reflected by the over-proportional increase of *svp*-expressing oCBs in *pnt* mutants first reported by (*Alvarez et al., 2003*). Our phenotypic analysis demonstrates that Edl is required for *svp* and *Doc* gene activity (the latter being due to restriction of *tin* expression) as well as the restriction of PntP2-dependent PntP1 expression in cardiac progenitors. Molecularly, Edl can modulate the activities of PntP2 as well as Yan (*Baker et al., 2001*; *Qiao et al., 2006*; *Qiao et al., 2004*; *Tootle et al., 2003*; *Vivekanand et al., 2004*; *Yamada et al., 2003*). The comparison of single and double mutant phenotypes, combined with the reproducibility of nearly all aspects of the cardiac *pnt* phenotype by Edl overexpression, implies that Edl acts primarily by inhibiting Pnt during cardiac cell diversification, although we cannot fully exclude additional interactions with Yan. Our observations further support the function of Edl as an antagonist of Pnt (first demonstrated in the context of eye and chordotonal organ development; *Yamada et al., 2003*) and rule out an initially proposed Pnt-stimulating function (*Baker et al., 2001*).

## Linkage of MAPK and Pnt activities

The involvement of Edl also leads to important conclusions regarding the placement of Pnt function within the cardiac gene regulatory network. Based on the phenotypic discrepancies between *pnt* and other EGF pathway components (gain and loss of CBs, respectively), Alvarez et al. proposed that PntP2 acts independent of MAPK signaling to limit the number of CBs (*Alvarez et al., 2003*). Since we found that Edl blocks Pnt activity in oCB progenitors, and Edl is thought to antagonize PntP2 mainly by blocking MAPK-dependent phosphorylation (*Qiao et al., 2006*), we propose that PntP2 acts downstream of MAPK also during cardiogenesis (see *Figure 10*). This is further supported by our data demonstrating *spi*-sensitive cardiac expression of PntP1 and the observation that, if timed properly, both EGF and Pnt activities can lead to expanded gCB and reduced oCB populations. However, not all MAPK activities require *pnt*, which is the case for the pro-cardiogenic activities of EGF. Notably, parallel *pnt*-dependent and *pnt*-independent MAPK signaling functions take place also during other processes such as epithelial branching morphogenesis (*Cabernard and Affolter, 2005*).

## Special features of Pnt-dependent regulation in working myocardial cells

Our model of CB diversification incorporates the observation that the PntP1 isoform is activated specifically in gCB progenitors in a PntP2-dependent and EGF-sensitive fashion. This is reminiscent of the situation in other tissues such as the developing eye where the PntP1 isoform is also activated in a MAPK/PntP2-dependent manner (*Gabay et al., 1996*; *O'Neill et al., 1994*; *Shwartz et al., 2013*). We propose that PntP1 becomes activated at a particular threshold of MAPK/PntP2 activity. This activation marks a point of no return for CB diversification, because PntP1 cannot be inhibited via Edl. The activation of PntP1 also explains why *edl* overexpression with relatively late acting drivers such as *tinD-GAL4* (as used in the *edl* mutant rescue experiment) does not cause the cardiac phenotypes observed with early pan-mesodermal drivers. Furthermore, depending on enhancer structure, target genes may be either quickly activated by PntP2 alone or require higher levels only achieved upon additional PntP1 buildup (particularly for sustained expression). In case of the *mid* gene, our model includes both possibilities (*Figure 10* and *Figure 10—figure supplement 1*). Although the exact details of this activation as well as the direct binding of Pnt to particular sites in vivo remain to

be investigated, the sum of our genetic and enhancer data provide strong indications for *mid* being a direct and functionally critical target of Pnt during cardiac cell diversification. Thus, by regulating Pnt activity, the timing of Mid protein appearance can be controlled. We predict that this timing is linked to its capability to interfere with *svp* expression, since later presence of Mid in all CB subtypes including oCBs (mediated by other, Pnt-independent mechanisms; see *Figure 9F,H,J*) does not lead to *svp* repression. One possible explanation for the co-occurrence of Svp and Mid in oCBs at later stages is that the chromatin structure determining *svp* gene activity becomes fixed prior to the delayed appearance of Mid protein in these cells.

Besides *pntP1* and *mid*, there are very likely additional target genes activated by PntP2 and/or PntP1 to execute the differentiation program in generic working myocardial cells. Incomplete conversion of gCBs in *mid* mutants also calls for the existence of additional repressors that contribute to oCB fate suppression. Interestingly, a study investigating Tin target genes found that cardiac target enhancers of Tin are not only enriched for Tin-binding sites but also for a motif highly reminiscent of ETS binding sites, termed 'cardiac enhancer enriched (CEE) motif' (with the consensus ATT[TG]CC or GG[CA]AAT in antisense orientation) (*Jin et al., 2013*). Mutation of four CEE sites (one of which overlapping our predicted ETS binding site) in a ca. 600 bp version of the *midE19* enhancer nearly abolished reporter activity in that study. Thus, many of the CEE-containing Tin target enhancers might in fact also be targets of Pnt (potentially mediating ETS-dependent activation) or Yan (potentially mediating ETS-dependent repression in the absence of MAPK signals). Therefore, a combination of closely spaced Tin and ETS binding sites might be a key signature in enhancers of working myocardial genes, although additional features must be present in their architecture to distinguish them from Tin+ETS binding site-containing enhancers active in pericardial cells or their progenitors (*Halfon et al., 2000*). The differences might include elements directly or indirectly regulated by Delta-Notch signaling. Notably, the juxtacrine Notch ligand Delta is upregulated in the CB lineage in an MAPK-activity-dependent manner (*Grigorian et al., 2011*). Hence, it is conceivable that Pnt proteins might stimulate *Delta* transcription in gCBs to control OPC development in a non-autonomous manner. This would explain both, simultaneous mis-specification of gCB progenitors and non-ostial-related OPCs in EGF mutants as well as phenotypic similarities between *pnt* mutants and mutants for components of the Delta-Notch signaling pathway. However, because of the herein described function of Pnt in suppressing *svp* transcription and oCB fate, *pnt* mutants feature an extreme bias in the increase of oCBs that has not been observed in Notch pathway mutants (*Albrecht et al., 2006*; this work).

## What is the original signal that discriminates generic and ostial progenitors?

Our work clearly identifies Pnt and Edl as crucial transducers of spatio-temporal inputs during cardiac cell diversification, but open questions remain regarding the initial source for the differential activities. Our model proposes that factors which tilt the balance between PntP2 activity and Edl will have a major impact on CB subtype choice (see *Figure 10*). Thus, any input that modestly increases MAPK/PntP2 activity within the appropriate window of time would favor gCB fate, whereas factors that have the opposite effect should promote oCB specification. This points to activities that impinge on the highly complex and dynamic expression of *rho* and/or *edl*. The Rhomboid protease is a key determinant in the decision of which cells will activate the more broadly expressed EGF Spitz and thus emanate signaling activity. A prime candidate for an instructive cue to anterior-posterior positioning within each segment could be Hh (indicated in the extended model in *Figure 10—figure supplement 1*), because it was proposed to be an oCB-promoting and *rho*/MAPK pathway-modulating signal towards the cardiogenic mesoderm in previous studies (*Liu et al., 2006*; *Ponzielli et al., 2002*). In these studies, decreased *svp* expression and reduced numbers of Tin-negative CBs observed in *hh* mutants and upon overexpression of constitutive repressor forms of the Hh effector Ci were interpreted as signs of Hh-dependent oCB specification, although no converse effects have been reported using constitutive active Ci forms. However, the role of the Hh pathway in CB diversification is not fully understood, mainly due to complications arising from ectodermal Hh functions, primarily in maintaining pro-cardiogenic *wg* expression (*Bejsovec and Martinez Arias, 1991*; *Park et al., 1996*). Furthermore, the effect of Hh on MAPK and *rho* activities in the dorsal mesoderm was suggested to be positive rather than negative based on an expansion of stage 12 mesodermal *rho* expression and expanded numbers of cells with activated MAPK upon pan-mesodermal overexpression of *hh* (*Liu et al., 2006*). This would refute a function favoring oCB fate, but it is an

interesting finding in light of our work, which couples *rho* activity with gCB specification. A modulation of *rho* expression via Hh signaling, whether direct or indirect, would also be consistent with the phenotype of mutants lacking the function of *patched* (encoding a negative regulator of Hh signaling activity), in which we observe a strong increase in the gCBs:oCBs ratio (although absolute CB numbers are highly variable between embryos and alleles; E. Heyland, F. Karama, B. Schwarz and I. Reim, unpublished observations). On the other hand, mutants with diminished Hh pathway activity, including some that were recovered by our EMS screen because of their partial CB losses (i.e. *smoothened* mutants), do not display a biased reduction of either oCBs or gCBs (E. Heyland, F. Karama, B. Schwarz and I. Reim; unpublished observations). Hence, the regulation of *rho* and the role of *hh* during CB diversification await more detailed analysis.

Factors that regulate *edl* expression levels might also determine the outcome of the competition between Edl and Pnt. The *edl* gene was found to be positively regulated by EGF signaling, and to be a target of Pnt and Yan, and thus was proposed to provide a negative feedback system for EGF inputs (*Baker et al., 2001*; *Leatherbarrow and Halfon, 2009*; *Vivekanand et al., 2004*; *Yamada et al., 2003*). Our extended model therefore includes regulation by Pnt as a possibility (dashed arrows in *Figure 10—figure supplement 1*). Nevertheless, additional or alternative inputs need to be considered to explain the strong *edl* expression in presumptive oCB progenitors with low Pnt activity. Notably, ChIP-on-chip experiments suggest that *edl* is also targeted by cardiogenic factors (*Junion et al., 2012*). Furthermore, *edl* was identified as a positively regulated target of Notch signaling in a *Drosophila* cell culture system (*Krejcí et al., 2009*). However, observed persistent *edl* expression in Notch pathway mutants argues against positive inputs from Notch during *edl* regulation in oCB progenitors.

The spatio-temporal dynamics and detailed mechanisms that regulate MAPK and *edl* activities within the cardiogenic mesoderm remain to be investigated in future studies. Such studies may also help to understand lineage decisions in other tissues and species. Edl/Mae-relatives are also present in non-Dipteran insects (e.g. *Tribolium*; *Bucher and Klingler, 2005*), echinoderms, and the chordate *Ciona*. Although no clear ortholog of Edl appears to be present in vertebrates, a SAM domain-only isoform of the human Yan-relative TEL2 as well as *Drosophila* Edl were shown to inhibit transcriptional stimulation by the mammalian Pnt orthologs ETS1/ETS2 in cell culture (*Gu et al., 2001*; *Vivekanand and Rebay, 2012*). Hence, the restriction of ETS protein activities by protein-protein interactions offers an intriguing mechanism to fine-tune MAPK signaling output in developing tissues of both invertebrates and vertebrates.

# Materials and methods

## Key resources table

| Reagent type (species) or resource | Designation | Source or reference | Identifiers | Additional information |
|---|---|---|---|---|
| Genetic reagent (*Drosophila melanogaster*) | S-18a-13b-16b.1 | PMID: 24935095 | | starter stock used for EMS mutagenesis; genotype: y[*] w[*]; P{RedH-Pelican.org-1-HN18-dsRed, w[+mC]}18a, P{pGD130.tinC*-GFP, y[+]}13b, P{RedH-Pelican.HLH54Fb-dsRed, w[+mC]}16b |
| Genetic reagent (*D. melanogaster*) | S-18a-13b-16c.1 | PMID: 24935095 | | starter stock used for EMS mutagenesis; genotype: y[*] w[*]; P{RedH-Pelican.org-1-HN18-dsRed, w[+mC]}18a, P{pGD130.tinC*-GFP, y[+]}13b, P{RedH-Pelican.HLH54Fb-dsRed, w[+mC]}16 c |
| Genetic reagent (*D. melanogaster*) | aop[1] | Bloomington Drosophila Stock Center | BDSC:3101 | |
| Genetic reagent (*D. melanogaster*) | bib[S1538] | this paper | | mutation in S-18a-13b-16c.1 background |
| Genetic reagent (*D. melanogaster*) | Df(2R)edl-S0520 | this paper | | mutation in S-18a-13b-16b.1 background |

*Continued on next page*

*Continued*

| Reagent type (species) or resource | Designation | Source or reference | Identifiers | Additional information |
|---|---|---|---|---|
| Genetic reagent (*D. melanogaster*) | edl[k06602] | Bloomington Drosophila Stock Center | BDSC:10633; FBal0057093 | |
| Genetic reagent (*D. melanogaster*) | Df(2R)edl-L19 | Y. Hiromi, PMID: 12874129 | FBab0037748 | |
| Genetic reagent (*D. melanogaster*) | P{edl.AF1}BS12; P{edl[+]} | this paper | | derived from injection with pCaSpeR4-X18C12-edl_rescue; line # BS12 carries P{edl.AF1} on chromosome 3 |
| Genetic reagent (*D. melanogaster*) | Egfr[f2] | Bloomington Drosophila Stock Center | BDSC:2768 | |
| Genetic reagent (*D. melanogaster*) | Egfr[S0167] | this paper | | mutation in S-18a-13b-16b.1 background |
| Genetic reagent (*D. melanogaster*) | Egfr[S2145] | this paper | | mutation in S-18a-13b-16c.1 background |
| Genetic reagent (*D. melanogaster*) | Egfr[S2307] | this paper | | mutation in S-18a-13b-16c.1 background |
| Genetic reagent (*D. melanogaster*) | Egfr[S2561] | this paper | | mutation in S-18a-13b-16c.1 background |
| Genetic reagent (*D. melanogaster*) | htl[YY262] | PMID: 8957001 | | |
| Genetic reagent (*D. melanogaster*) | mam[S0669] | this paper | | mutation in S-18a-13b-16b.1 background |
| Genetic reagent (*D. melanogaster*) | mam[S4648] | this paper | | mutation in S-18a-13b-16c.1 background |
| Genetic reagent (*D. melanogaster*) | mid[1] | Bloomington Drosophila Stock Center | BDSC:3086 | |
| Genetic reagent (*D. melanogaster*) | mid[S0021] | this paper | | mutation in S-18a-13b-16b.1 background |
| Genetic reagent (*D. melanogaster*) | midE19-GFP | M. Frasch; PMID: 23326246 | | |
| Genetic reagent (*D. melanogaster*) | mid180-GFP | this paper | | insertion in attP2 |
| Genetic reagent (*D. melanogaster*) | mid180-mETS-GFP | this paper | | insertion in attP2 |
| Genetic reagent (*D. melanogaster*) | pnt[MI03880] | Bloomington Drosophila Stock Center | BDSC:37615 | |
| Genetic reagent (*D. melanogaster*) | pnt[Δ88] | Bloomington Drosophila Stock Center | BDSC:861 | |
| Genetic reagent (*D. melanogaster*) | pyr[18] | PMID: 19515694 | | |
| Genetic reagent (*D. melanogaster*) | pyr[S3547] | PMID: 22609944 | | |
| Genetic reagent (*D. melanogaster*) | rho[7M43] | Bloomington Drosophila Stock Center | BDSC:1471 | |
| Genetic reagent (*D. melanogaster*) | rho[L68] | Bloomington Drosophila Stock Center | BDSC:9095 | |
| Genetic reagent (*D. melanogaster*) | S[S4550] | this paper | | mutation in S-18a-13b-16c.1 background |
| Genetic reagent (*D. melanogaster*) | S[B0453] | F. Schnorrer; PMID: 18327265 | | |
| Genetic reagent (*D. melanogaster*) | spi[S3384] | this paper | | mutation in S-18a-13b-16c.1 background |

*Continued on next page*

*Continued*

| Reagent type (species) or resource | Designation | Source or reference | Identifiers | Additional information |
|---|---|---|---|---|
| Genetic reagent (D. melanogaster) | spi[1] | Bloomington Drosophila Stock Center | BDSC:1859; FBal0016005 | |
| Genetic reagent (D. melanogaster) | svp[AE127]-lacZ | Y. Hiromi, PMID: 11404079 | | |
| Genetic reagent (D. melanogaster) | ths[759] | PMID: 19515694 | | |
| Genetic reagent (D. melanogaster) | 'tin-ABD;tin[EC40]' | PMID: 16987868 | | |
| Genetic reagent (D. melanogaster) | UAS-aop.ACT-IIa | Bloomington Drosophila Stock Center | BDSC:5789 | |
| Genetic reagent (D. melanogaster) | UAS-edl-X | Y. Hiromi, PMID: 12874129 | | |
| Genetic reagent (D. melanogaster) | 'UAS-Egfr[DN].B-29-77-1; UAS-EgfrDN.B-29-8-1'; 2x EGFR[DN] | Bloomington Drosophila Stock Center | BDSC:5364 | |
| Genetic reagent (D. melanogaster) | UAS-mid-B2 | PMID: 15922573 | | |
| Genetic reagent (D. melanogaster) | UAS-pntP1-3 | Bloomington Drosophila Stock Center | BDSC:869 | |
| Genetic reagent (D. melanogaster) | UAS-pntP2[VP16]—2 | C. Klämbt; PMID: 11051548 | | |
| Genetic reagent (D. melanogaster) | UAS-p35 | Bloomington Drosophila Stock Center | BDSC:5073 | |
| Genetic reagent (D. melanogaster) | UAS-rho[EP3704] | Bloomington Drosophila Stock Center | BDSC:17276 | |
| Genetic reagent (D. melanogaster) | UAS-rho(ve.dC) | Bloomington Drosophila Stock Center | BDSC:8858 | |
| Genetic reagent (D. melanogaster) | UAS-svp | M. Hoch | | |
| Genetic reagent (D. melanogaster) | how[24B]-GAL4; 24B | Bloomington Drosophila Stock Center | BDSC:1767 | |
| Genetic reagent (D. melanogaster) | tinCΔ4-GAL4 | M. Frasch; PMID: 11404079 | | |
| Genetic reagent (D. melanogaster) | tinD-GAL4 | J. Weiss; PMID: 16221729 | | |
| Genetic reagent (D. melanogaster) | 2xPE-twi-GAL4 | Bloomington Drosophila Stock Center | BDSC:2517 | |
| Genetic reagent (D. melanogaster) | Df(2L)Exel6006 | Bloomington Drosophila Stock Center | BDSC:8000 | |
| Genetic reagent (D. melanogaster) | Df(2R)BSC25 | Bloomington Drosophila Stock Center | BDSC:6865 | |
| Genetic reagent (D. melanogaster) | Df(2R)Exel7157 | Bloomington Drosophila Stock Center | BDSC:7894 | |
| Genetic reagent (D. melanogaster) | Df(3R)Exel9012 | Bloomington Drosophila Stock Center | BDSC:7990 | |
| Genetic reagent (D. melanogaster) | lbe-GFP | Bloomington Drosophila Stock Center | BDSC:55822 | |
| Genetic reagent (D. melanogaster) | pnt-GFP | Bloomington Drosophila Stock Center | BDSC:42680 | |
| Recombinant DNA reagent | pCaSpeR4-X18C12-edl_rescue (plasmid) | Y. Hiromi, PMID: 12874129 | | P transformation plasmid for generation of P{edl.AF1} |

*Continued on next page*

*Continued*

| Reagent type (species) or resource | Designation | Source or reference | Identifiers | Additional information |
|---|---|---|---|---|
| Antibody | anti-Doc2+3 (guinea pig polyclonal) | PMID: 12783790 | | (1:2000, TSA) |
| Antibody | anti-Doc3+2 (guinea pig polyclonal) | PMID: 12783790 | | (1:1000) |
| Antibody | anti-H15 (rabbit polyclonal) | J. Skeath; PMID: 19013145 | | (1:2000) |
| Antibody | anti-H15 (guinea pig polyclonal) | J. Skeath; PMID: 19013145 | | (1:2000) |
| Antibody | anti-Mid (rabbit polyclonal) | J. Skeath; PMID: 19013145 | | (1:250, TSA or 1:1000) |
| Antibody | anti-PntP1 (rabbit polyclonal) | J. Skeath; PMID: 12756183 | | (1:250) |
| Antibody | anti-Mef2 (rabbit polyclonal) | H.T. Nguyen | | (1:1500) |
| Antibody | anti-Odd (rat polyclonal) | PMID: 9683745 | | (1:600, TSA) |
| Antibody | anti-Eve (rabbit polyclonal) | PMID: 2884106 | | (1:3000) |
| Antibody | anti-Tin (rabbit polyclonal) | PMID: 9362473 | | (1:750) |
| Antibody | anti-Zfh1 (rabbit polyclonal) | R. Lehmann; PMID: 9435286 | | (1:2000) |
| Antibody | anti-dpMAPK (mouse monoclonal) | Sigma | | (1:500, TSA) |
| Antibody | anti-Seven-up 5B11 (mouse monoclonal) | Developmental Studies Hybridoma Bank | | (1:20, TSA) |
| Antibody | anti-Wg 4D4 (mouse monoclonal) | Developmental Studies Hybridoma Bank | | (1:30, TSA) |
| Antibody | anti-β-galactosidase 40-1a (mouse monoclonal) | Developmental Studies Hybridoma Bank | | (1:50, TSA or 1:20) |
| Antibody | anti-β-galactosidase (rabbit polyclonal) | Cappel | | (1:1500) |
| Antibody | anti-GFP (rabbit polyclonal) | Molecular Probes | Molecular Probes:A6455 | (1:2000) |
| Antibody | anti-GFP (rabbit polyclonal) | Rockland | Biomol:600-401-215 | (1:1000) |
| Antibody | anti-GFP 3E6 (mouse monoclonal) | Life Technologies | Life Technologies:A11120 | (1:100, TSA) |
| Antibody | anti-cleaved-Caspase-3 Asp175 (rabbit polyclonal) | Cell Signaling Technology | Cell Signaling Technology:#9661 | (1:100, TSA) |
| Antibody | sheep anti-Digoxigenin (sheep polyclonal) | Roche | Roche:11333089001 | (1:1000, TSA) |
| Commercial assay or kit | VectaStain Elite ABC-HRP kit | Vector Laboratories | Linaris:PK-6100 | |
| Commercial assay or kit | tyramide signal amplification (TSA) reagent Cy3 | PerkinElmer | PerkinElmer: SAT704A001EA | |
| Commercial assay or kit | tyramide signal amplification (TSA) reagent Fluorescein | PerkinElmer | PerkinElmer: SAT701001EA | |
| Commercial assay or kit | TUNEL apoptosis detection kit (Apoptag) | Millipore | Millipore:S7100 | |

## *Drosophila melanogaster* stocks

The mutants *bib^S1538^*, *Df(2R)edl-S0520*, *Egfr^S0167^*, *Egfr^S2145^*, *Egfr^S2307^*, *Egfr^S2561^*, *kuz^S3330^*, *kuz^S3832^*, *mam^S0669^*, *mam^S4648^*, *mid^S0021^*, *mid^S2961^*, *numb^S1342^*, *numb^S3992^*, *numb^S4439^*, *pyr^S3547^* (*Reim et al., 2012*), *spi^S3384^*, *Star^S4550^* were recovered from our EMS screen. The lines *mid^1^*, *UAS-mid-B2*, *how^24B^-GAL4*, *pnr^MD237^-GAL4*, *svp^AE127^-lacZ* (a *svp* mutant in homozygous condition), *UAS-svp.l*, *2xPE-twi-GAL4*, *twi-SG24-GAL4*, *tinD-GAL4*, *UAS-tin#2*, *{tin-ABD}T003-1B1*; *tin^EC40^*, *UAS-p35* were as described previously (*Reim et al., 2012*; *Reim et al., 2005*; *Zaffran et al., 2006*). In addition, the following strains were used: *aop^1^ = aop^IP^* (*Nüsslein-Volhard et al., 1984*; *Rogge et al., 1995*), *UAS-aop.ACT-IIa* (*Rebay and Rubin, 1995*), *bib^1^* (*Lehmann et al., 1983*), *edl^L19^ = Df(2R)edl-L19* (*edl* and some neighboring genes deleted) and *UAS-edl-X* (both from Y. Hiromi; *Yamada et al., 2003*), *P{lacW}edl^k06602^* (*Baker et al., 2001*; *Török et al., 1993*), *Egfr^f2^* (*Clifford and Schüpbach, 1994*), *UAS-Egfr^DN^.B-29-77-1;UAS-Egfr^DN^.B-29-8-1* (*Buff et al., 1998*), *htl^YY262^* (*Gisselbrecht et al., 1996*), *kuz^e29-4^* (*Rooke et al., 1996*), *PBac{lbe-GFP.FPTB}VK00037* (A. Victorsen and K. White), *mam^8^* (*Lehmann et al., 1983*), *mid^1^* (*Buescher et al., 2004*), *midE19-GFP* (*Jin et al., 2013*; from M. Frasch), *pnt^Δ88^* (*Scholz et al., 1993*), *pnt^MI03880^* (PntP2-specific; harbors a gene-trap cassette with an artificial splice acceptor followed by stop codons upstream of the *pntP1* transcription start site; *Venken et al., 2011*), *UAS-pntP2^VP16^-2* (*Halfon et al., 2000*; originally from C. Klämbt), *UAS-pntP1-3* and *UAS-pntP2-2* (*Klaes et al., 1994*), *PBac{pnt-GFP.FPTB}VK00037* (R. Spokony and K. White; *Boisclair Lachance et al., 2014*), *pyr^18^* and *ths^759^* (*Klingseisen et al., 2009*), *rho^7M43^* (*Jürgens et al., 1984*), *rho^L68^* (*Salzberg et al., 1994*), *rho^EP3704^* (*Bidet et al., 2003*), *UAS-rho(ve.dC)* (*de Celis et al., 1997*), *spi^1^ = spiIIA^IIA14^* (*Nüsslein-Volhard et al., 1984*), *Star^B0453^* (*Chen et al., 2008*; from F. Schnorrer), *tinCΔ4-GAL4* (*Lo and Frasch, 2001*; from M. Frasch), *Df(2R)Exel7157*, and about 180 additional deficiencies spanning chromosome 2 (except where noted, all stocks available from the Bloomington Stock Center).

Flies expressing *edl^+^* from a transgene were generated anew by standard P-element transgenesis using the previously described *edl[+t18]* rescue construct (named AF1 in *Yamada et al., 2003*; provided by Y. Hiromi). Line *P{edl.AF1}BS12* carrying an insertion on chromosome three was used in this study.

Unless noted otherwise, *y w* or *S-18a-13b-16c.1* control (*Hollfelder et al., 2014*) flies were used as wild type controls. Mutant lines were maintained over *GFP-* or *lacZ*-containing balancer chromosomes to allow recognition of homozygous embryos. Flies were raised at 25°C, except for UAS/GAL4-driven overexpression at 29°C.

## Isolation and mapping of novel EMS mutants

Novel EMS-induced mutants were obtained from our screen for embryonic heart and muscle defects and mapped to a particular gene through extensive complementation testing analogous to the previously described procedure (*Hollfelder et al., 2014*). Many alleles were mapped by unbiased complementation tests with a set of chromosome 2 deficiencies and subsequent non-complementation of lethality and embryonic phenotype by previously described alleles. *Df(2R)edl-S0520* was mapped by non-complementation of lethality with *Df(2R)Exel7157*, *Df(2R)edl-L19* and *Df(2R)ED3636*, but the cardiac phenotype was only reproduced in trans with *Df(2R)Exel7157*, *Df(2R)edl-L19* and *edl^k06602^*. Novel alleles of *Egfr* and *Star* were mapped using a candidate gene approach.

## Molecular analysis of mutations and deletions

Several EMS alleles and the unmutagenized *S-18a-13b-16c.1* control were analyzed by sequencing of overlapping PCR products covering the coding sequence and splicing sites of the candidate gene as described (*Hollfelder et al., 2014*). Details about the mutations are provided in *Supplementary file 1*-Table S1. The area deleted by *Df(2R)edl-S0520* and its approximate break points were determined by iterative PCR amplification tests. The insertion of *P{lacW}edl^k06602^* near the *edl* transcription start site was confirmed by PCR using primers binding to the 5' *P* end and adjacent genomic DNA. Although the integrity of the both *P* element ends could be confirmed by PCR, no genomic *edl* sequences expected next to the 3' *P* end could be amplified using several primer pairs shown to amplify control DNA. This indicates that *P{lacW}edl^k06602^* is associated with a deletion in *edl*. Details of the deletion mapping are listed in *Supplementary file 2*-Table S2.

## Generation of reporter constructs for enhancer analysis

The *mid180-GFP* reporter constructs were generated according to a similar *lacZ* construct published by *Ryu et al. (2011)*. The forward primer 5'-*Eco*RI-CGTGCCTCCCACTTCAGGGCGG-3' and the backward primer 5'-*Bam*HI-TTAATTTCATTTTTCACTCTGCTCACTTGAGATTCCCCTGCTTTGTCTGCGGC**ATT*TCC*G**CTTCT-3' were used to amply DNA from *y w* flies. The predicted ETS binding site matching the antisense sequence of published ETS binding motifs (*Halfon et al., 2000*; *Hollenhorst et al., 2011*; underlined) was mutated in *mid180-mETS-GFP* by replacing the invariable *TCC* core (bold) with *AAA* in the backward primer. Amplicons were cloned into *Eco*RI/*Bam*HI of pH-Stinger-attB (*Jin et al., 2013*), sequenced and inserted into the *attP2* landing site via *nos*-driven ΦC31 integrase.

## Staining procedures

Embryo fixations, immunostainings for proteins and RNA in situ hybridizations were carried out essentially as described (*Knirr et al., 1999*; *Reim and Frasch, 2005*), except for stainings with anti-dpMAPK, for which the formaldehyde concentration was doubled and embryos were rehydrated from methanol and stained immediately after fixation. VectaStain Elite ABC-HRP kit (Vector Laboratories) and tyramide signal amplification (TSA, PerkinElmer Inc.) were used for detection of RNA and certain antigens (as indicated). The following antibodies were used: guinea pig anti-Doc2+3 (1:2000, TSA) and anti-Doc3+2 (1:1000) (*Reim et al., 2003*), rabbit anti-H15/Nmr1 (1:2000), guinea pig anti-H15/Nmr1 (1:2000), rabbit anti-Mid/Nmr2 (early stages: 1:250, TSA; late stages: 1:1000 direct) and rabbit anti-PntP1 (1:250, TSA) (all from J. Skeath; *Alvarez et al., 2003*; *Leal et al., 2009*), rabbit anti-Mef2 (1:1500) (from H.T. Nguyen), rat anti-Odd (1:600, TSA) (*Kosman et al., 1998*), rabbit anti-Eve (1:3000) (*Frasch et al., 1987*), rabbit anti-Tin (1:750) (*Yin et al., 1997*) (all from M. Frasch), rabbit anti-Zfh1 (1:2000) (from R. Lehmann; *Broihier et al., 1998*), mouse anti-dpMAPK (Sigma, 1:500, TSA), rabbit anti-β-galactosidase (Cappel, 1:1500), rabbit anti-GFP (Molecular Probes, 1:2000 and Rockland, 1:1000), mouse anti-GFP 3E6 (Life Technologies, 1:100, TSA), anti-cleaved-Caspase-3 (Asp175, Cell Signaling Technology, 1:100, TSA), sheep anti-Digoxigenin (Roche, 1:1000, TSA), monoclonal mouse antibodies anti-β-galactosidase 40-1a (1:20 direct or 1:50 with TSA), anti-Seven-up 5B11 (1:20, TSA) and anti-Wg 4D4 (1:30, TSA) (all from Developmental Studies Hybridoma Bank, University of Iowa), fluorescent secondary antibodies (1:200) (Jackson ImmunoResearch Laboratories and Abcam), biotinylated secondary antibodies (1:500) and HRP-conjugated anti-rabbit and anti-mouse IgG (1:1000) (Vector Laboratories). TUNEL staining was performed as described (*Reim et al., 2003*) using the Millipore ApopTag S7100 kit in combination with TSA.

Digoxigenin-labeled antisense riboprobes against *mid, edl, rho* and *pntP2* were used for whole mount in situ hybridizations. The *mid* probe was generated as described previously (*Reim et al., 2005*). T7 promoter-tagged *edl, rho* and *pntP2* (isoform-specific exons) templates for in vitro transcription were generated by PCR (primers *edl*: CAATCGTGAAAGAGCGAGGGTC, T7-TGACGAGCAGAACTAAGGACTAGGC, *edl^intron*: GCACCGACGACTCAACTTCCTG, T7-GCTGCGATTGCGATTACAAACAAG, *pnt*: CCAGCAGCCACCTCAATTCGGTC, T7-GCGTGCGTCTCGTTGGGGTAATTG, *rho*: ATGGAGAACTTAACGCAGAATGTAAACG, T7-TTAGGACACTCCCAGGTCG) from DNA of wild-type flies or flies carrying *UAS-rho(ve.dC)* or *UAS-pntP2*, respectively.

Embryos were mounted in Vectashield (Vector Laboratories). Images were acquired on a Leica SP5 II confocal laser scanning microscope and projected using Leica LAS-AF and ImageJ.

## Acknowledgements

We are grateful to Manfred Frasch for critical reading of the manuscript, Patrick Lo and Christoph Schaub for their contributions to the EMS screen, Edmar Heyland, Emi Vargatoth, Tanja Drechsler and Angela Bruns for technical assistance. We thank Manfred Frasch, Yasushi Hiromi, James Skeath, Hanh Nguyen, Frank Schnorrer, the Bloomington Stock Center and the Developmental Studies Hybridoma Bank (University of Iowa) for providing fly stocks or reagents.

## Additional information

### Funding

| Funder | Grant reference number | Author |
|---|---|---|
| Deutsche Forschungsge-meinschaft | RE 2985/1-1 | Ingolf Reim |

The funders had no role in study design, data collection and interpretation, or the decision to submit the work for publication.

### Author contributions

Benjamin Schwarz, Conceptualization, Data curation, Formal analysis, Validation, Investigation, Methodology, Writing—original draft, Writing—review and editing; Dominik Hollfelder, Data curation, Investigation, Methodology; Katharina Scharf, Leonie Hartmann, Formal analysis, Investigation, Visualization; Ingolf Reim, Conceptualization, Data curation, Formal analysis, Supervision, Funding acquisition, Investigation, Visualization, Methodology, Writing—original draft, Project administration, Writing—review and editing

### Author ORCIDs

Ingolf Reim http://orcid.org/0000-0001-8069-5532

### Decision letter and Author response

Decision letter https://doi.org/10.7554/eLife.32847.034
Author response https://doi.org/10.7554/eLife.32847.035

## Additional files

### Supplementary files

• Supplementary file 1. Table S1. Alleles with cardioblast patterning defects isolated and/or characterized in this study. The table lists the results from the genetic, phenotypic and molecular analysis of the characterized mutants. Indicated nucleotide positions are relative to transcription start site of transcript RA and amino acid positions of protein isoform PA (FB2017_01, released February 14, 2017; D. melanogaster R6.14); * indicates a nonsense mutation, n.d.: not determined.
DOI: https://doi.org/10.7554/eLife.32847.030

• Supplementary file 2. Table S2. Characterization of *edl* deletions via PCR. Presence (+) or absence (-) of DNA fragments after PCR reaction including genomic DNA from homozygous *S-18a-13b-16c.1* control (*WT*), *Df(2R)edl-S0520*, *edl^{k06602}* or *Df(2R)edl-L19* animals and primer pairs as indicated. CDS: part of coding sequence, TSS: transcription start site, n.d.: not determined. Amplicons are listed in linear order as located on chromosome 2R. * Six additional intronic *GEFmeso* amplicons were also negative in *S0520*.
DOI: https://doi.org/10.7554/eLife.32847.031

• Transparent reporting form
DOI: https://doi.org/10.7554/eLife.32847.032

### Data availability

All data generated or analysed during this study are included in the manuscript and supporting files. Source data files have been provided for Figures 1, 3, 5 and 5-S1.

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
