## [Decision Letter]

Thank you for sending your article entitled "Diversification of heart progenitor cells by EGF signaling and differential modulation of ETS protein activity" for peer review at *eLife*. Your article has been favorably evaluated by K VijayRaghavan (Senior Editor) and three reviewers, one of whom, Utpal Banerjee, is a member of our Board of Reviewing Editors. The following individual involved in review of your submission has agreed to reveal his identity: Rolf Bodmer (Reviewer #3).

The revisions asked for by the reviewers are rather extensive and in the post-review discussion, the concern was raised that they are not achievable in a two-month period, and even that some of the concerns may be difficult to meet to the satisfaction of an *eLife* publication. Nevertheless, the reviewing editor finds considerable merit and novelty in the manuscript given that this is an area of *Drosophila* developmental biology that needs attention provided by the manuscript.

Summary:

In this paper, the authors show that EGFR signaling has a specific and defined role in the specification of cardioblast cell fates. Their work shows that active EGFR signaling through the Spitz signaling involving Rhomboid and Star activate the EGFR signaling in the gCB cells and is required for the activation and maintenance of Tinman expression. This process requires the participation of both isoforms of Pointed and also requires the function of Tbx20 ortholog, Midline. In contrast, the specification of the oCB requires an active attenuation of EGFR signaling and this is mediated by the expression of the gene ETS domain lacking (Edl) which has been shown to bind to ETS domain protein and block their function.

Using loss of function and gain of function studies the authors convincingly support their model for the role of EGFR signaling in the specification of cardioblast fate during heart development in *Drosophila*.

Essential revisions:

1) In the paper the authors refer to the fact that EGFR signaling functions at multiple stages, one being its role in the specification of cardioblast fate. They use loss of function mutants and overexpression and dominant negative version driven by broadly expressed GAL4 lines to come of the conclusion. The results will be strengthened if EGFR signaling is blocked at specific time points to rule out the possibility that a prior EGFR step might contribute to the phenotype.

2) The authors should show the activity status of EGFR during these fate specification events by using the activated MAPK antibody to define how this correlates with the phenotype.

3) Even though this paper is directed towards defining the role of EGFR in cardioblast fate, the role of Notch signaling during these events have been totally ignored. Is it possible that Notch signaling is required for the activation of Edl, which then blocks the activation of Pnt mediated transcription to promote oCB fate?

4) Although the authors suggest that the decrease in gCBs is due to the adoption of other cell fates, they do not identify which fates these cells take on. A better understanding of this through staining with markers for other cell types, clonal analysis or lineage tracing would be helpful, especially to distinguish this model from one in which mutant cells undergo apoptosis or fail to divide.

5) While the final figure shows a brief enhancer analysis, this aspect of the paper is under-developed. This point is specifically germane to Figure 7, for which there are a number of concerns:

A) There is no indication that *pointed* can bind specifically to the ETS sites in the enhancer, and that its binding is abrogated by the mutations.

B) Since the role for midline in cardiac diversification is expected to occur during stage 12, it is unclear why the enhancer analyses (panels D and E) are presented at stage 16.

C) Related to this, the *midE19* enhancer is clearly not reporting the entire midline expression pattern, so how generally applicable are the findings from the analysis of just this single enhancer?

D) Since midline is expressed at later stages throughout the cardiac tube, how does it fail to suppress *seven-up* at later stages (the authors discuss this later in the paper, but a mechanistic understanding is not apparent)?

6) How does hedgehog impact EGF expression and receptor activity? This is a concern because, as the work stands, we are not left with a comprehensive understanding of how cardioblast diversification is triggered that incorporates existing published data.

7) The question of spatial integration (in addition to temporal integration that is addressed in this paper), attributing the spatial cues to *wg* and in particular *hh* signaling, upstream of *rho* needs better elaboration. Also, the Discussion should be more explicit on the fact that the current studies mainly address a permissive/transducing role for *pnt/edl* rather than a spatially instructive role, and/or discuss what the scenarios could be that relate to spatially instructive signals.

8) Also, the gCBs come in two flavors Tin^+^/Lbe^+^ and Tin^+^ only; their distinction should be included in the Discussion.

---

## [Author Response]

Essential revisions:1) In the paper the authors refer to the fact that EGFR signaling functions at multiple stages, one being its role in the specification of cardioblast fate. They use loss of function mutants and overexpression and dominant negative version driven by broadly expressed GAL4 lines to come of the conclusion. The results will be strengthened if EGFR signaling is blocked at specific time points to rule out the possibility that a prior EGFR step might contribute to the phenotype.

First, we want to clarify that the manuscript proposes multiple functions of EGF signaling in the cardiogenic mesoderm only for gCB progenitor selection as such (Pnt-independent function) as well as for a potential boost in MAPK activity that feeds into CB subtype diversification (Pnt-dependent function). Our experiment that uses dominant negative EGFR^DN^ driven by *2xPE-twi+how^24B^-GAL4* (Figure 1J and the corresponding quantification in Figure 1M) was merely intended to demonstrate that EGFR signaling activity is required in the mesoderm and exclude indirect influences from potential disruptions in the ectoderm in mutants. The discussed mesodermal EGF signaling activities might not be fully separable by blocking EGFR at specific time points with conventional methods due to potential temporal overlap as well as technical reasons discussed below.

Secondly, we would like to point out that there are no indications for an early pan-mesodermal function of EGF signaling. In the early mesoderm, MAPK activity is strictly dependent on FGF signaling (Michelson, Gisselbrecht, Buff, and Skeath, 1998; Wilson, Vogelsang and Leptin, 2005; Klingseisen, Clark, Gryzik, and Müller, 2009), and in the fully migrated mesoderm, earliest MAPK activity is observed in the FGF-dependent Eve^+^ C2 cluster, slightly before MAPK activity in the DA1-generating C15 cluster and the nearly simultaneously appearing cardiogenic clusters C14/C16 (Carmena, Gisselbrecht, Harrison, Jiménez, and Michelson, 1998; Grigorian, Mandal, Hakimi, Ortiz, and Hartenstein, 2011). While gCBs and subsets of PCs are strongly affected in amorphic *Star* and *rhomboid* mutants as well as in *2xPE-twi+how^24B^-GAL4>>EGFR^DN^* embryos, oCBs, oCB-related OPCs and Eve-positive pericardial cells are not (Figure 1, Figure 3), demonstrating that mesodermal EGFR activities are clearly specific to particular cardiac lineages. This and the normal early patterning of the *rho* expression clusters in the cardiogenic mesoderm themselves (newly added results in Figure 2) also argue against an earlier broad function. The specificity is analogous to published data in certain lineages in the somatic mesoderm, where EGF signaling, via loss-of-function approaches similar to ours, was shown to affect only certain types of muscles (e.g. muscle DA1; Buff, Carmena, Gisselbrecht, Jiménez, and Michelson, 1998; Carmena, Gisselbrecht, Harrison, Jiménez, and Michelson, 1998) and adult muscle precursors (AMPs; Figeac, Jagla, Aradhya, Da Ponte, and Jagla, 2010). These functional aspects of EGF signaling in the mesoderm are now mentioned more specifically in the Introduction.

Furthermore, we think that the experiment shown in Figure 1J actually represents a partial and rather late attenuation of EGF signaling due to delays in the GAL4 system in combination with the need to accumulate EGFR^DN^ at levels capable of out-competing endogenous EGFR. We have tried to block EGF signaling using EGFR^DN^ driven by the later active *tinD+tinCΔ4-GAL4* driver, but this did not result in any abnormalities in CB formation. Since formation of the EGF-dependent DA1 muscle was also entirely normal in this background, we conclude that this driver does not produce sufficient amounts of dominant-negative EGFR^DN^ for EGF signaling attenuation at the time of cardioblast specification. However, a mild but significant reduction in gCBs was observed using *24B-GAL4* without *2xPE-twi-GAL4* (data not shown; average number of gCBs=72.2, p=0.0049*; oCBs: 27.7, p=0.85; n=15), even though this driver was still too weak to generate any DA1 defects (n=3). Aiming at a more efficient, controllable knock-down of EGFR activity we have tried to knock-down Rhomboid, the most restricted factor of EGFR signaling, via inducible RNAi. Unfortunately, this turned out to be unsuccessful, presumably because tissue-specific RNAi is generally very inefficient during embryonic development of *Drosophila* (even the use of *2xPE-twi+how^24B^-GAL4* in combination with three different UAS-RNAi lines did not influence heart or DA1 muscle formation; data not shown). Note that effective expression of positively acting factors in the pathway (overexpression of Rhomboid, PntP1) is inherently easier to achieve, since there is no competition with the endogenous counterpart. This (together with artificial activation of normally FGF-regulated MAPK upon early *rho* expression) explains the differences in the efficiency of the different drivers in loss- and gain-of-function experiments.

2) The authors should show the activity status of EGFR during these fate specification events by using the activated MAPK antibody to define how this correlates with the phenotype.

To clarify the EGFR status in the cardiogenic mesoderm, we have performed stainings with anti-diphospho-MAPK antibodies in combination with cardiogenic mesoderm markers in wild-type embryos and compared them to stainings of various EGF-related genotypes. The most important results of this analysis are shown in Figure 2. The data are consistent with the assertion that EGF signaling is the major source of MAPK activity in CB progenitors. (FGF signaling is assumed to contribute to the residual dpMAPK activity observed in EGF pathway mutants, particularly after CB specification (late stage 12/stage 13). This is in agreement with a CB fate maintaining function of FGF signaling proposed by Grigorian et al. (Grigorian, Mandal, Hakimi, Ortiz, and Hartenstein, 2011.)

Concerning activated MAPK level differences between CB subtypes: These are technically difficult to prove by standard dpMAPK antibody stainings. Furthermore, differential input to the different progenitor subtypes may not solely rely on level differences, but could also be achieved via the timing/duration of activity. Thus to get a full understanding, activity would have to be monitored for each cell type over the entire time course of early cardiac development. However, since MAPK activation is expected to be upstream of cardioblast diversification events, there is no marker available that will unambiguously identify oCB vs. gCB progenitors in the early cardiogenic mesoderm, making it impossible to assign dpMAPK antibody signals with 100% certainty to particular CB progenitor subtypes at critical stages.

3) Even though this paper is directed towards defining the role of EGFR in cardioblast fate, the role of Notch signaling during these events have been totally ignored. Is it possible that Notch signaling is required for the activation of Edl, which then blocks the activation of Pnt mediated transcription to promote oCB fate?

The revised manuscript now more extensively deals with Notch-related aspects (see added text in the Results section related to Figure 5, corresponding supplemental figures and discussion in distinction to *pnt* phenotypes). Several observations argue against a positive oCB-subtype determining function of Notch signaling.

1) Genetic data show that Notch signaling activity generally antagonizes CB fate.

From our mutagenesis screen, we also obtained several mutants for genes required for Notch signaling, i.e. *mastermind (mam), big brain (bib)* and *kuzbanian (kuz)*. We did not include these mutants in the first version of the manuscript, because Notch signaling components, particularly *mam* and *kuz*, have been previously reported to cause supernumerary CBs irrespective of their CB subtype (Hartenstein, Rugendorff, Tepass and Hartenstein, 1992; Tao, Christiansen and Schulz, 2007, and in particular the CB subtype analysis in Albrecht, Wang, Holz, Bergter, and Paululat, 2006). In response to your request, we added a new figure supplement showing cardioblast patterning data for various Notch signaling-impaired genotypes including a novel molecularly characterized *bib* allele (Figure 5—figure supplement 1). Consistent with the previous reports, our data show that Notch signaling mutants feature a strong increase in both oCB and gCB numbers. Some variations in the oCB:gCB ratio are expected to occur between different Notch-related gene mutations or alleles due to variable impact (e.g. due to different maternal contributions or allele strength) on the multiple functions of Notch, which involve lateral inhibition and fate decisions linked to asymmetric division, the latter being relevant specifically for the oCB lineage. In case of the novel *mam* allele, *mam^S0669^*, the oCB:gCB ratio was elevated relative to the wild type, but not as much as in *pnt* mutants. By contrast, in mutants with a strong maternal component such as *kuz*, the increase in oCBs lags a bit behind that of gCBs presumably due to partially functional lateral inhibition during selection of the oCB progenitors (Albrecht, Wang, Holz, Bergter, and Paululat, 2006; and our own observations). Many Notch-related mutations may cause oCB:gCB ratios that lie somewhere in between (mam^S0669^, bib^S1538^

; a normal oCB:gCB ratio was also observed upon mesoderm-specific overexpression of dominant-negative MamN driven by *how^24B^-GAL4*, which featured a milder increase in total cardioblast numbers; Jan Wittstatt and Ingolf Reim, unpublished data). Of note, hyperactivation of the Notch pathway via constitutive active Notch driven by *tinCΔ4-GAL4* or *how^24B^-GAL4* causes a complete or near complete loss of cardioblasts. In those cases where some cardioblasts are formed, the fraction of oCBs was not increased compared to the wild type (Jan Wittstatt and Ingolf Reim, unpublished data).

2) Cardiac *edl* expression is not abolished in Notch pathway mutants.

Nevertheless, because a previous report identified *edl* as a Notch target in a *Drosophila* cell culture system (Krejčí, Bernard, Housden, Collins, and Bray, 2009), it made sense to test whether this is the case in our system as well. Thus, we analyzed *edl* expression (using an intronic probe to get a higher spatio-temporal resolution) in wild type embryos (new Figure 4—figure supplement 2) and compared it to embryos with impaired Notch activity (*mam* mutant analysis shown in the new Figure 5—figure supplement 2 and similar data for *bib* not shown). These mutants do not display any obvious reductions of *edl* expression in the cardiogenic mesoderm, further arguing against a substantial role of Notch in determining CB subtypes.

4) Although the authors suggest that the decrease in gCBs is due to the adoption of other cell fates, they do not identify which fates these cells take on. A better understanding of this through staining with markers for other cell types, clonal analysis or lineage tracing would be helpful, especially to distinguish this model from one in which mutant cells undergo apoptosis or fail to divide.

The sum of our data suggests that missing gCBs do not simply differentiate into one particular alternate cell type in mutants with diminished EGF signaling. At the end of embryogenesis, Tin^+^ and Tin^-^(Odd^+^) PCs are not expanded in compensation for gCBs, but are rather reduced (Figure 1—figure supplement 2, Figure 3, Figure 3—figure supplement 1) as are many somatic muscles (Buff, Carmena, Gisselbrecht, Jiménez, and Michelson, 1998, and our own data not shown). We discuss that a minor fraction of cells may undergo apoptosis or adopt oCB fates. But these effects are small and they do not fully explain the fate of all lost progenitors.

To follow up on the fate of the unaccounted cells, we analyzed *Star* mutants carrying a *pnr-lacZ* reporter that marks the definitive cardiogenic mesoderm (excluding EPC progenitors) and early CBs and OPCs (as well as dorsal ectoderm and amnioserosa). Although these experiments tended to favor the idea that the presumptive gCBs cells become part of the pool of uncommitted *Mef2*^+^/Lmd^+^ dorsal mesoderm cells (frequent observance of LacZ^+^/*Mef2*^+^ or LacZ^+^/Lmd^+^ cells at the dorsal margin of the mesoderm), results were not fully conclusive due to insufficient specificity of the marker. Even in the wild type, low level expression was also detected in a few *Mef2*^+^/Lmd^+^ cells in the vicinity of developing abdominal CBs/OPCs and in a broader dorsal mesoderm region in thoracic segments; later stages also displayed expression in passing hemocytes. The somewhat broader expression of the marker in the wild type is actually consistent with the idea that some more cells obtain the competence to develop into cardiac cells than will be selected by opposing MAPK and Notch activities. Thus, if some pro-cardiogenic cells fail to get properly selected upon attenuation of EGF signaling they will not be discriminable from other cells of the region. A potential increase in the number of unselected pro-cardiogenic cells is technically difficult to prove since this pool is poorly defined and the number of missing progenitors is relatively small in comparison to the remaining dorsal mesoderm cells.

We modified wording in the respective paragraphs of the Results and Discussion sections to acknowledge the uncertainties regarding the fate of presumptive gCBs.

5) While the final figure shows a brief enhancer analysis, this aspect of the paper is under-developed. This point is specifically germane to Figure 7, for which there are a number of concerns:

*A) There is no indication that* pointed *can bind specifically to the ETS sites in the enhancer, and that its binding is abrogated by the mutations.*

Although we agree that there is currently no evidence for direct binding of Pnt to the ETS sites in the *midE19* enhancer, we have provided several strong indications for this:

First, the *midE19* enhancer responds to the absence of Pnt with a very severe drop in activity.

Second, upon ectopic Pnt expression the enhancer gains activity even in Tin-negative cardioblasts.

Third, the minimal enhancer version, *mid180*, loses activity upon mutation of a single motif that matches the well-characterized ETS binding signature (as compiled from plenty of in vitro as well as in vivo data in Hollenhorst, McIntosh, and Graves, 2011; see also Halfon, Carmena, Gisselbrecht, Sackerson, Jiménez, and Baylies, 2000).

Due to the sequence match with the well-established ETS binding signature, we think that additional in vitro binding studies would yield results that are entirely predictable and thus would not be helpful in generating new insights. Therefore we tried to detect binding in vivo by chromatin immunoprecipitation (ChIP) using anti-PntP1 (which may represent only a small fraction of bound Pnt) or anti-GFP with chromatin from C-terminally tagged Pnt::GFP embryos. However, enrichment was inconsistent for the ETS site-containing region of *mid180* and never observed for the *eve* mesoderm enhancer. Since the latter is considered a well-established Pnt target (although no Pnt ChIP data are available), we attribute this to technical difficulties that are likely to be caused by the following obstacles: (1) gCB progenitors only represent a very small number of cells in the context of whole embryos, (2) Pnt is likely to bind many target genes in more abundant cell types, and (3) active Pnt may occupy targets only for a short time.

Nevertheless, we are still confident about the Pnt-*midE19* interaction on the basis of our genetic data. We would like to emphasize that even in the unlikely case that Pnt only indirectly regulates *mid* expression or via a different enhancer, the regulatory logic in our model (now more in the focus in the simplified model version) would still be maintained. We acknowledge the fact that direct binding has not been validated in the Discussion.

B) Since the role for midline in cardiac diversification is expected to occur during stage 12, it is unclear why the enhancer analyses (panels D and E) are presented at stage 16.

There is a delay in the detectability of the GFP reporter and we find the current representation with the straight alignment of bilateral cardioblast rows more instructive. We have now expanded the previous Figure 7 by including reporter analysis in embryos at earlier stages (in part also stained for GFP RNA instead of GFP protein to enable earlier detection). For a more suitable presentation we decided to split this expanded figure into two separate main figures (Figure 8 and Figure 9). Simultaneously the former Figure 7—figure supplement 1 has been embedded into the main figure.

A notable observation is that the less robust *mid180-GFP* always shows identical signal strength in both sibling gCBs derived from a common progenitor, which implies that the level of *mid180* enhancer activity is already established within the progenitor even though reporter expression is only detectable after this progenitor divides. The small, but noticeable delay with which this enhancer kicks in in comparison to *midE19* and to endogenous activation could be due to loss of some auxiliary binding sites which would raise the threshold at which Pnt leads to enhancer activity (e.g., out of its genomic context, the minimal *mid180* version could be "on hold" until PntP2 has activated sufficient PntP1).

C) Related to this, the midE19 enhancer is clearly not reporting the entire midline expression pattern, so how generally applicable are the findings from the analysis of just this single enhancer?

The pattern displayed by the *midE19*-driven reporter gene reflects the alternating pattern of early gCB-specific *mid* expression and it is responsive to Pnt activity just like the endogenous *mid* gene during germ band retraction stages (old Figure 7G, I; now Figure 9F, H). The observation that this enhancer does not drive gene expression in oCBs after germ band retraction as detected for *mid* in the genomic context (old Figure 7J, now Figure 9I), suggests that additional enhancers or interaction with other *cis*-regulatory regions are required to reproduce all aspects of cardiac *mid* expression. This is now stated more clearly in the manuscript. What causes the later uniform expression of *mid* in all cardioblasts is outside of the topic of this work as it is regulated by *pnt*-independent, yet unknown mechanisms (old Figure 7K, now Figure 9J).

D) Since midline is expressed at later stages throughout the cardiac tube, how does it fail to suppress seven-up at later stages (the authors discuss this later in the paper, but a mechanistic understanding is not apparent)?

One possible explanation is that the chromatin structure determining *svp* gene activity becomes fixed prior to appearance of Mid protein in oCBs. This is consistent with the observation that *svp* activation requires Tin (Ryan, Hendren, Helander, and Cripps, 2007), which is present in oCB progenitors but absent from maturing oCBs. We added this proposed explanation to the Discussion.

6) How does hedgehog impact EGF expression and receptor activity? This is a concern because, as the work stands, we are not left with a comprehensive understanding of how cardioblast diversification is triggered that incorporates existing published data.

See joint response to comments 6+7 below.

7) The question of spatial integration (in addition to temporal integration that is addressed in this paper), attributing the spatial cues to wg and in particular hh signaling, upstream of rho needs better elaboration. Also, the Discussion should be more explicit on the fact that the current studies mainly address a permissive/transducing role for pnt/edl rather than a spatially instructive role, and/or discuss what the scenarios could be that relate to spatially instructive signals.

To make the issue of spatial inputs more apparent and also to address the second part of this comment we have made modifications to the discussion, referring more extensively to published data (particularly from Liu, Qian, Wessells, Bidet, Jagla, and Bodmer, 2006) and mentioning some of our own unpublished observations regarding components of the Hh pathway in the final paragraph of the Discussion. Our current data are not in agreement with the view that the Hh pathway directly promotes oCB fate/*svp* expression (Ponzielli, Astier, Chartier, Gallet, Thérond, and Sémériva, 2002). Mutants with diminished Hh pathway activity, including some that were recovered by our EMS screen because of their partial CB losses (i.e. *smo* mutants), do not display a biased reduction of oCBs (E. Heyland, F. Karama, B. Schwarz and I. Reim; unpublished observations). On the other hand, loss of *patched*, which encodes a negative regulator of Hh signaling activity and therefore would be expected to favor oCBs, leads to a relative and in some alleles absolute reduction of oCBs in comparison gCBs, although absolute CB numbers are highly variable between embryos and alleles (E. Heyland, F. Karama, B. Schwarz and I. Reim; unpublished observations; Tao, Christiansen and Schulz, 2007). In the presumed amorphic mutant *ptc^9^* we found an increase in CB numbers that was solely based on supernumerary gCBs. Although this may also be is caused indirectly via changes in the ectoderm, this observation is in agreement with the previously reported positive input of Hh signaling on *rho* expression (Liu, Qian, Wessells, Bidet, Jagla, and Bodmer, 2006) and the positive impact of *rho* on gCBs described in our manuscript. Therefore, we added inputs from Hh to the current working model (in the extended version), but labeled it with a question mark for unknown details. Since potentially specific effects of Hh signaling may be masked by interaction with ectodermal Wg signaling or other feedback mechanisms, this is not a matter of a few-months investigation, but a full-scale project. We are currently setting up such a project, because we agree with the critique that differences in Pnt activities, whether promoted by MAPK signaling or attenuated by Edl ultimately must be connected to segmental inputs.

8) Also, the gCBs come in two flavors Tin^+^/Lbe^+^ and Tin^+^ only; their distinction should be included in the Discussion.

In response to the reviewers request, we paid more attention to the distinction of Tin^+^/Lbe^+^ and Tin^+^/Lbe^-^ gCB subtypes and analyzed the frequency of their retention in mutants with impaired EGF signaling. Although we did not detect a significant bias in the retention/loss of a particular (abdominal) gCB subtype, this analysis provided some valuable additional information, which we added to the manuscript (Figure 1—figure supplement 3). We found that residual gCBs usually occur either as Lbe^+^ or Lbe^-^ pairs, but rarely as mixed Lbe^+^/Lbe^-^ pairs or single cells. This further supports our notion that EGF function is required prior to completion of the final mitotic division at the gCB progenitor stage.